# Breaking the Communication-Privacy-Accuracy Trilemma

**Wei-Ning Chen**
Department of Electrical Engineering
Stanford University
wnchen@stanford.edu

**Peter Kairouz**
Google
kairouz@google.com

**Ayfer Özgür**
Department of Electrical Engineering
Stanford University
aozgur@stanford.edu

## Abstract

Two major challenges in distributed learning and estimation are 1) preserving the privacy of the local samples; and 2) communicating them efficiently to a central server, while achieving high accuracy for the end-to-end task. While there has been significant interest in addressing each of these challenges separately in the recent literature, treatments that simultaneously address both challenges are still largely missing. In this paper, we develop novel encoding and decoding mechanisms that simultaneously achieve optimal privacy and communication efficiency in various canonical settings. In particular, we consider the problems of mean estimation and frequency estimation under $\varepsilon$-local differential privacy and $b$-bit communication constraints. For mean estimation, we propose a scheme based on Kashin's representation and random sampling, with order-optimal estimation error under both constraints. For frequency estimation, we present a mechanism that leverages the recursive structure of Walsh-Hadamard matrices and achieves order-optimal estimation error for *all* privacy levels and communication budgets. As a by-product, we also construct a distribution estimation mechanism that is rate-optimal for all privacy regimes and communication constraints, extending recent work that is limited to $b = 1$ and $\varepsilon = O(1)$. Our results demonstrate that intelligent encoding under joint privacy and communication constraints can yield a performance that matches the optimal accuracy achievable under either constraint alone.

## 1 Introduction

The rapid growth of large-scale datasets has been stimulating interest in and demands for distributed learning and estimation, where datasets are often too large and too sensitive to be stored on a centralized machine. When data is distributed across multiple devices, communication cost often becomes a bottleneck of modern machine learning tasks [38]. This is even more so in federated learning type settings, where communication occurs over bandwidth-limited wireless links [31]. Moreover, as more personal data is entrusted to data aggregators, in many applications it carries sensitive individual information, and hence finding ways to protect individual privacy is of crucial importance. In particular, local differential privacy (LDP) [18, 21, 34, 49] is a widely adopted privacy paradigm, which guarantees that the outcome from a privatization mechanism will not release too much individual information statistically. In this paper, we study the relationship between utility (often in forms of accuracy for certain statistical tasks), privacy, and communication *jointly*.

At first glance, privacy and communication may seem to be in conflict with each other: achieving privacy requires the addition of noise, therefore increasing the entropy of the data and making it less compressible. For instance, consider the mean estimation problem, which appears as a fundamental subroutine in many distributed optimization tasks, e.g. distributed stochastic gradient descent (SGD). Here, the goal is to estimate the empirical mean of a collection of $d$-dimensional vectors. If we first privatize each vector via `PrivUnit` in [13] (which is optimal under LDP constraints) and then quantize via the `RandomSampling` quantizer in [24] (which is optimal under communication constrains), a tedious but straightforward calculation shows that the resulting $\ell_2$ estimation error grows with $d^2$. However, this is far from matching the error rate under each constraint separately, which has a linear dependence on $d$. A similar phenomenon happens in the distribution estimation problem, where each client's data is drawn independently from a discrete distribution $\boldsymbol{p}$ with support size $d$. One can satisfy both constraints by first perturbing the data via the Subset Selection (SS) mechanism [51] (which is optimal under LDP constraints) and then quantizing the noised data to $b$ bits. Again, it can be shown that under such strategy, the $\ell_2$ estimation error of $\boldsymbol{p}$ has a quadratic dependence on $d$. This leaves a huge gap to the lower bounds under each constraint separately, which have a linear dependence on $d$. See Section A in the appendix for a detailed discussion.

While there has been significant recent progress on understanding how to achieve optimal accuracy under separate privacy [12, 51] and communication [42, 52] constraints, as illustrated above a simple concatenated application of these optimal schemes can yield a highly suboptimal performance. Recent works that attempt to break this communication-privacy-accuracy trilemma have been either limited to specific regimes or, as we show, are far from optimal. For example, [3] provides a 1-bit $\varepsilon$-LDP scheme for distribution estimation which is order-optimal only in the low communication regime ($b = O(1)$) and high privacy regime ($\varepsilon = O(1)$), while [24] tries to address both constraints in the mean estimation setting, but the error rate achieved under their mechanism is quadratic in $d$ and therefore does not improve on the above baseline. We note that the general privacy regime (i.e. $\varepsilon = \Omega(1)$) is also of both theoretical and practical interest. For instance, when $n = \Omega(d)$, one can combine LDP with amplification techniques [7, 19, 20] to ensure stronger central differential privacy.

This paper closes the above gaps *for any given privacy level $\varepsilon$ and communication budget $b$*. Indeed, our results show that the fundamental trade-offs are determined by the more stringent of the two constraints, and with careful encoding we can satisfy the less stringent constraint *for free*, thus breaking the privacy-communication-accuracy trilemma. For the same privacy level $\varepsilon$, this allows us to achieve the accuracy of existing mechanisms in the literature with drastically smaller communication budget, or equivalently, for the same communication budget achieve higher privacy. It also explains, for example, why 1-bit communication budget is sufficient under the high privacy regime [3, 11]. We will demonstrate this phenomenon in various canonical tasks and answer the following question: *"given arbitrary privacy budget $\varepsilon$ and communication budget $b$, what are the fundamental limits for estimation accuracy?"* We next formally define the settings and the problem formulations we consider in this paper.

## 1.1 Problem Formulation

The general distributed statistical tasks we consider in this paper can be formulated as follows: each one of the $n$ clients has local data $X_i \in \mathcal{X}$ and sends a message $Y_i \in \mathcal{Y}$ to the server, who upon receiving $Y^n$ aims to estimate some pre-specified quantity of $X^n$. Note that $X^n$ are *not necessarily drawn from some distribution*. At client $i$, the message $Y_i$ is generated via some mechanism (a randomized mapping that possibly uses shared randomness across participating clients and the server) denoted by a conditional probability $Q_i(y|X_i)$ satisfying the following constraints.

**Local differential privacy**    Let $(\mathcal{Y}, \mathcal{B})$ be a measurable space, and $Q(\cdot|x)$ be probability measures for all $x \in \mathcal{X}$, with $\{Q(\cdot|x)|x \in \mathcal{X}\}$ dominated by some $\sigma$-finite measure $\mu$ so that the density $Q(y|x)$ exists. A mechanism $Q$ is $\varepsilon$-LDP if

$$\forall x, x' \in \mathcal{X}, \, y \in \mathcal{Y}, \, \frac{Q(y|x)}{Q(y|x')} \le e^\varepsilon.$$

**$b$-bit communication constraint**    $\mathcal{Y}$ satisfies $b$-bit communication constraint if each of its elements can be described by $b$ bits, i.e. $|\mathcal{Y}| \le 2^b$.

The goal is to jointly design a mechanism (at clients' sides) and an estimator (at the server side) so that the accuracy of estimating some target function $\sum_{i=1}^{n} f(X_i)$ is maximized. In this paper, we are mainly interested in the *distribution-free* framework, that is, we do not assume any underlying distribution on $X_i$, but we also demonstrate that our results can be extended to probabilistic settings. To this end, we will focus on the following four canonical tasks.

**Mean estimation**   For real-valued data, we consider the $d$-dimensional unit euclidean ball $\mathcal{X} = \mathcal{B}_d(\mathbf{0}, 1)$ and are interested in estimating the *empirical mean* $\bar{X} \triangleq \frac{1}{n}\sum_i X_i$. The goal is to minimize the worst-case $\ell_2$ estimation error defined as

$$r_{\text{ME}}(\ell_2, \varepsilon, b) \triangleq \min_{(\hat{X}, Q^n)} \max_{X^n \in \mathcal{X}^n} \mathbb{E}\left[\left\|\hat{X} - \bar{X}\right\|_2^2\right], \qquad (1)$$

where $Q^n$ satisfies $\varepsilon$-LDP and $b$-bit communication constraints. When the context is clear, we may omit $\varepsilon$ and $b$ in $r_{\text{ME}}(\ell, \varepsilon, b)$.

**Statistical mean estimation**   In the probabilistic version of the mean estimation problem, we assume that $X_i$'s are drawn from some common but unknown distribution $P$ supported on $\mathbf{B}_d(\mathbf{0}, 1)$, the goal is to estimate the *statistical mean* $\theta(P) = \mathbb{E}_P[X_1]$ and to minimize the $\ell_2$ estimation error:

$$r_{\text{SME}}(\ell_2, \varepsilon, b) \triangleq \min_{(\hat{\theta}, Q^n)} \max_{X^n \in \mathcal{X}^n} \mathbb{E}\left[\left\|\hat{\theta}(X^n) - \theta(P)\right\|_2^2\right].$$

**Frequency estimation**   When $\mathcal{X}$ consists of categorical data, i.e. $\mathcal{X} = [d] = \{1, ..., d\}$, we are interested in estimating $D_{X^n}(x) \triangleq \frac{1}{n}\sum_i \mathbb{1}_{\{X_i = x\}}$ for $x \in [d]$. With a slight abuse of notation, $D_{X^n}$ is viewed as a vector $(D_{X^n}(1), ..., D_{X^n}(d))$ lying in the $d$-dimensional probability simplex. The worst-case estimation error is defined by

$$r_{\text{FE}}(\ell, \varepsilon, b) \triangleq \min_{(\hat{D}, Q^n)} \max_{X^n \in \mathcal{X}^n} \mathbb{E}\left[\ell\left(\hat{D}, D_{X^n}\right)\right],$$

where $\ell = \|\cdot\|_\infty, \|\cdot\|_1$, or $\|\cdot\|_2^2$ and again $Q^n$ satisfies $\varepsilon$-LDP and $b$-bit communication constraints.

**Distribution estimation**   A closely related setting is that of discrete distribution estimation, where we assume that the $X_i$'s are drawn independently from a discrete distribution $\mathbf{p}$ on the alphabet $\mathcal{X} = [d]$, and the goal is to estimate $\mathbf{p}$. In this case, the worst-case error is given by

$$r_{\text{DE}}(\ell, \varepsilon, b) \triangleq \inf_{(Q^n, \hat{\mathbf{p}})} \sup_{\mathbf{p} \in \mathcal{P}_d} \mathbb{E}[\ell(\hat{\mathbf{p}}, \mathbf{p})],$$

where $\mathcal{P}_d$ is the $d$-dimensional probability simplex.

We note that these canonical tasks serve as fundamental subroutines in many distributed optimization and learning problems. For instance, the convergence rate of distributed SGD is determined by the $\ell_2$ error of estimating the mean of the local gradient vectors (see [5] for more on this connection). Lloyd's algorithm [35] for k-means clustering or the power-iteration method for PCA can also be reduced to the mean estimation task.

**Remark 1.1** *In this work, we generally assume the availability of shared randomness across the participating clients and the server. In this case the encoding functions at each node can be explicitly denoted as $Q_i(y|X_i, U)$ where $U$ is a shared random variable that is independent of data, referred to as a public coin. $U$ is also available at the server and the estimator implicitly depends on $U$. In our notation, we suppress this dependence on $U$ for simplicity. The entropy of $U$ is referred as the amount of shared randomness needed by a scheme. In Section B, we discuss the amount of shared randomness required by our schemes in order to achieve the optimal estimation error. We point out that in the statistical settings (i.e. statistical mean estimation and distribution estimation), the optimal estimation error can be achieved without shared randomness.*

|  | Privacy | Comm. | $\ell_2$ error |
|---|---|---|---|
| SQKR (this work, Thm. 2.1) | $\forall \varepsilon$ | $\forall b$ | $\frac{d}{n \min(\varepsilon^2, \varepsilon, b)}$ |
| Cross-polytope [24] | $\varepsilon \succeq 1$ | $b \succeq \log d$ | $\frac{d^2}{n}$ |
| Simplex [24] | $\varepsilon \succeq \log d$ | $b \succeq \log d$ | $\frac{d}{n}$ |

Table 1: Comparison between our mean estimation scheme and vqSGD [24]. Our scheme applies to general communication and privacy regimes, and achieves optimal estimation error for all scenarios.

## 1.2   Relation to Prior Work

Previous works in the mean estimation problem [6, 10, 24, 42, 44, 50] mainly focus on reducing communication cost, for instance, by random rotation [42] and sparsification [6, 14, 48, 50]. Among them, [24] considers LDP simultaneously. It proposes vector quantization and takes privacy into account, developing a scheme for $\varepsilon = \Theta(1)$ and $b = \Theta(\log d)$ with estimation error $O(d^2/n)$. In contrast, the scheme we develop in Theorem 2.1 achieves an estimation error $O(d/n)$ when $\varepsilon = \Theta(1)$ and $b = \Theta(\log d)$. Moreover, our scheme is applicable for any $\varepsilon$ and $b$ and achieves the optimal estimation error, which we show by proving a matching information theoretic lower bound. See Table 1 for a comparison of our results with [24]. A key step in our scheme is to pre-process the local data via Kashin's representation [36]. While various compression schemes, based on quantization, sparsification and dithering have been proposed in the recent literature and Kashin's representation for communication efficiency [16, 23, 40, 41] or for LDP [22] has been also explored in a few works , it is particularly powerful in the case of joint communication and privacy constraints as it helps spread the information in a vector evenly in every dimension. This helps mitigate the error due to subsequent noise introduced by privatization and compression.

The recent works of [37, 47] also consider estimating empirical mean under $\varepsilon$-LDP. They show that if the data is from a $d$-dimensional unit $\ell_\infty$ ball, i.e. $X_i \in [-1, 1]^d$, then directly quantizing, sampling and perturbing each entry can achieve optimal $\ell_\infty$ estimation error that matches the LDP lower bound in [17], where their privatization steps are based on techniques developed in [13, 17]. Nevertheless, their approach does not yield good $\ell_2$ error in general. Indeed, as in the case of separation schemes discussed in Section A, the $\ell_2$ error of their scheme can grow with $d^2$. We emphasize that in many applications the $\ell_2$ estimation error (i.e. MSE) is a more appropriate measure than $\ell_\infty$. For instance, [5] shows a direct connection between the MSE in mean estimation and the convergence rate of distributed SGD.

Frequency estimation under local differential privacy has been studied in [46], where they propose schemes for estimating the frequency of an individual symbol and minimizing the variance of the estimator. Some of their schemes, while matching the information-theoretic lower bound on $\ell_2$ estimation error under privacy constraints, require large communication. For instance, the scheme Optimal Unary Encoding (OUE), which can be viewed as an asymmetric version of RAPPOR [53], achieves optimal $\ell_2$ estimation error, but the communication required is $O(d)$ bits, which, as we show in this work, can be reduced to $O(\min(\lceil \varepsilon \rceil, \log d))$ bits. We do this by developing a new scheme for

|  | Loss | Estimation error | Communication |
|---|---|---|---|
| Asymmertic RAPPOR [46, 53] | $\ell_2$ | $\Theta\left(\frac{d}{n \min\left((e^\varepsilon - 1)^2, e^\varepsilon\right)}\right)$ | $d$ bits |
| RHR (this work, Thm 3.1) | $\ell_2$ | $\Theta\left(\frac{d}{n \min\left((e^\varepsilon - 1)^2, e^\varepsilon\right)}\right)$ | $\min\left(\lceil \varepsilon \rceil, \log d\right)$ bits |
| Heavy hitter (Thm. 3.1 and [12]) | $\ell_\infty$ | $\Theta\left(\sqrt{\frac{\log d}{n \min(\varepsilon, \varepsilon^2)}}\right)$ | $\lceil \varepsilon \rceil$ bits |

Table 2: Comparison of different frequency estimation schemes.

| Privacy | $\varepsilon \in (0,1)$ | $\varepsilon \in (1, \log d)$ |
|---|---|---|
| SS [51] | $d$ bits | $\max\left(\frac{d}{e^{\varepsilon}}, \log d\right)$ |
| HR [4] | $\log d$ bits | $\log d$ bits |
| 1bit-HR [3] | 1 bit | - |
| RHR (this work, Thm. 3.2) | 1 bit | $\min\left(\lceil \varepsilon \rceil, \log d\right)$ |

Table 3: Comparison between LDP distribution estimation schemes, where blue(or red) color indicates that accuracy of the corresponding scheme is optimal (or not). Under same privacy guarantee, our scheme is more communication efficient while achieves same accuracy.

frequency estimation under joint privacy and communication constraints. We establish the optimality of our proposed schemes by deriving matching information theoretic lower bounds on $r_{\mathsf{FE}}\left(\ell_2, \varepsilon, b\right)$.

Frequency estimation is also closely related to heavy hitter estimation [3, 11, 12, 15, 29, 39, 53], where the goal is to discover symbols that appear frequently in a given data set and estimate their frequencies. This can be done if the error of estimating the frequency of each individual symbol can be controlled uniformly (i.e. by a common bound), and thus is equivalent to minimizing the $\ell_{\infty}$ error of estimated frequencies, i.e. $r_{\mathsf{FE}}\left(\ell_{\infty}, \varepsilon, b\right)$. It is shown in [12] that in the high privacy regime $\varepsilon = O(1)$, $r_{\mathsf{FE}}\left(\ell_{\infty}, \varepsilon, b\right) = \Theta(\sqrt{\log d / n \varepsilon^2})$, and this rate can be achieved via a 1-bit public-coin scheme that has a runtime almost linear in $n$ [11]. An extension, which we describe in Section E.4 of the appendix, generalizes the achievability in [12] to arbitrary $\varepsilon$ and $b$, achieving $r_{\mathsf{FE}}\left(\ell_{\infty}, \varepsilon, b\right) = O(\sqrt{\log d / n \min\left(\varepsilon^2, \varepsilon, b\right)})$. We compare our scheme and existing results in Table 2.

If we further assume $X^n$ are drawn from some discrete distribution $\boldsymbol{p}$, then the problem falls into distribution estimation under local differential privacy [1–4, 17, 30, 45, 51, 53] and limited communication [1, 2, 9, 14, 25, 27, 28, 52]. Tight lower bounds are given separately: for instance [4, 51] shows $r_{\mathsf{DE}}\left(\ell_1, \varepsilon, \log d\right) = \Omega(\sqrt{d^2/n \min((e^{\varepsilon}-1)^2, e^{\varepsilon})})$ and [27] shows $r_{\mathsf{DE}}\left(\ell_1, \infty, b\right) = \Omega(\sqrt{d^2/n 2^b})$.

We show that these lower bounds can be achieved simultaneously (Theorem 3.2). Our result recovers the result of [3] when $b = 1$ and $\varepsilon = O(1)$ as a special case. See Table 3 for a comparison.

### 1.3 Our Contributions and Techniques

To summarize, our main technical contributions include:

- For mean estimation, we characterize the optimal $\ell_2$ error $r_{\mathsf{ME}}\left(\ell_2\right) = \Theta\left(d/n \min\left(\varepsilon^2, \varepsilon, b\right)\right)$, by designing a public-coin scheme, Subsampled and Quantized Kashin's Response (SQKR), and proving its optimality by deriving matching information theoretic bounds (in Theorem 2.1). Our encoding scheme is based on Kashin's representation [36] and random sampling, which allow the server to construct unbiased estimator of each $X_i$ privately and with little communication. This significantly improves on [24], which focuses on the special case $\varepsilon = \Omega(1), b = \log d$ and achieves quadratic dependence on $d$ in that case.

- For frequency estimation, we characterize the optimal $\ell_1$ and $\ell_2$ errors under both constraints (in Theorem 3.1) and propose an order-optimal public-coin scheme called Recursive Hadamard Response (RHR). Our result shows that the accuracy is dominated only by the worst-case constraint, and this implies that one can achieve the less stringent constraint for free. The proposed scheme RHR is based on Hadamard transform, but unlike previous works using Hadamard transform, e.g. [11], we crucially leverage the recursive structure of the Hadamard matrix, which allows us to make the estimation error decay exponentially as $\varepsilon$ and $b$ grow. RHR is computationally efficient, and the decoding complexity is $O(n + d \log d)$. We establish its optimality by showing matching lower bounds on the performance.

- We show that RHR easily leads to an optimal scheme for distribution estimation [3, 4, 51], in which case it does not require shared randomness and achieves order-optimal $\ell_1$ and $\ell_2$ error for all

privacy regimes and communication budgets. We also provide empirical evidence that our scheme requires significantly less communication while achieving the same accuracy and privacy levels as the state-of-the-art approaches. See Section C for more results.

## 2 Mean Estimation

In the mean estimation problem, each client has a $d$-dimensional vector $X_i$ from the Euclidean unit ball, and the goal is to estimate the empirical mean $\bar{X} = \frac{1}{n}\sum_i X_i$ under $\varepsilon$-LDP and $b$ bits communication constraints. This problem has applications in private and communication efficient distributed SGD. The following theorem characterizes the optimal $\ell_2$ estimation error for this setting.

**Theorem 2.1** *For mean estimation under $\varepsilon$-LDP and $b$-bit communication constraints, we can achieve*

$$r_{\text{ME}}\left(\ell_2, \varepsilon, b\right) \preceq d/n \min\left(\varepsilon^2, \varepsilon, b\right). \tag{2}$$

*Moreover, if* $\min(\varepsilon^2, \varepsilon, b) = o(d)$ *and* $n \cdot \min(\varepsilon^2, \varepsilon, b) > d$, *the above error is optimal.*

Note that by taking $\varepsilon \to \infty$ for a fixed $b$, or by taking $b \to \infty$ for a fixed $\varepsilon$ in part (i), Theorem 2.1 provides the optimal error when we have the corresponding constraint alone. Furthermore, for finite $\varepsilon$ and $b$ we see that the optimal error is dictated by the error due to one of these constraints, the one that leads to larger error, and hence the less stringent constraint is satisfied for free. This also implies that to achieve the optimal accuracy under $\varepsilon$-LDP constraints, we do not need more than $\lceil\varepsilon\rceil$ bits. We note that the two conditions for optimality in the theorem are standard and are needed to restrict the problem to the interesting parameter regime.

The lower bounds are obtained by connecting the problem to a specific parametric estimation problem with a distribution supported on the unit ball. To match this lower bound, we propose a public-coin scheme, Subsampled and Quantized Kashin's Response (SQKR), based on Kashin's representation [36] and random sampling.

### 2.1 Subsampled and Quantized Kashin's Response

For each observation $X_i$, we aim to construct an unbiased estimator $\hat{X}_i$ which is $\varepsilon$-LDP, can be described in $b$ bits, and has small variance. Towards this goal, our general strategy is to quantize, subsample, and privatize the data $X_i$. However before this, it is crucial to pre-process each $X_i$ by a carefully designed mechanism to increase the robustness of the signal to noise introduced by sampling and privatization.

**Pre-processing via Kashin's representation** We first introduce the idea of a tight frame in Kashin's representation. A tight frame is a set of vectors $\{u_j\}_{j=1}^N \in \mathbb{R}^d$ that satisfy Parseval's identity, i.e. $\|x\|_2^2 = \sum_{j=1}^N \langle u_j, x\rangle^2$ for all $x \in \mathbb{R}^d$. A frame can be viewed as a generalization of the notion of an orthogonal basis in $\mathbb{R}^d$ for $N > d$. To increase robustness, we wish the information to be spread evenly across different coefficients. Thus, we say that the expansion $x = \sum_{j=1}^N a_j u_j$ is a Kashin's representation of $x$ at level $K$ if $\max_j |a_j| \leq \frac{K}{\sqrt{N}}\|x\|_2$ [33]. [36] shows that if $N > (1+\mu)d$ for some $\mu > 0$, then there exists a tight frame $\{u_j\}_{j=1}^N$ such that for any $x \in \mathbb{R}^d$, one can find a Kashin's representation at level $K = \Theta(1)$. This implies that we can represent each $X_i$ with coefficients $\{a_j\}_{j=1}^N \in [-c/\sqrt{d}, c/\sqrt{d}]^{c'd}$ for some constants $c$ and $c'$.

**Quantization** Each client $i$ computes the Kashin's representation $\{a_j\}_{j=1}^N \in [-c/\sqrt{d}, c/\sqrt{d}]^{c'd}$ of $X_i$, and then quantizes each $a_j$ into a 1-bit message $q_j \in \left\{-c/\sqrt{d}, c/\sqrt{d}\right\}$ with $\mathbb{E}[q_j] = a_j$. This yields an unbiased estimator of $\{a_j\}_{j=1}^N$, which can be described in $\Theta(d)$ bits in total. Moreover, due to the small range of each $a_j$, the variance of $q_j$ is bounded by $O(1/d)$.

**Sampling and privatization** To further reduce $\{q_j\}$ to $k = \min(\lceil\epsilon\rceil, b)$ bits, client $i$ draws $k$ independent samples from $\{q_j\}_{j=1}^N$ with the help of shared randomness, and privatizes its $k$ bits

message via $2^k$-RR mechanism [32, 49], yielding the final privatized report of $k$ bits, which it sends to the server.

Upon receiving the report from client $i$, the server can construct unbiased estimators $\hat{a}_j$ for each $\{a_j\}_{j=1}^N$, and hence reconstruct $\hat{X}_i = \sum_{j=1}^N \hat{a}_j u_j$, which yields an unbiased estimator of $X_i$. We show that the variance of $\hat{X}_i$ can be controlled by $O\left(d/\min\left(\varepsilon^2, \varepsilon, b\right)\right)$. Therefore $\frac{1}{n}\sum_i \hat{X}_i$ achieves the order-optimal $\ell_2$ estimation error, establishing the upper bound in Theorem 2.1. We provide a detailed description of the scheme and its performance analysis in Section D.

**Remark 2.1** *In order to achieve optimal communication efficiency, SQKR uses public randomness at the sampling step. That being said, we can still turn SQKR into a private scheme by using additional communication. See Section B for more details.*

At a high-level, SQKR resembles vqSGD [24] as both schemes seek a suitably designed representation for $X_i$ before quantizing it. vqSGD represents $X_i$ by a basis $B = \{b_1, ..., b_K\} \subset \mathbb{R}^d$ where $B$ is chosen in such a way that its convex hull contains the unit $\ell_2$ ball. Therefore we can write $X_i = \sum_{j=1}^N a_j b_j$ with $\sum_j a_j = 1$. Equivalently, the pre-processing step of vqSGD corresponds to a linear transformation that embeds the $d$-dim $\ell_2$ unit ball into a $N$-dim $\ell_1$ ball. In contrast, Kashin's representation above embeds the $d$-dim $\ell_2$ unit ball into an $N$-dim $\ell_\infty$ ball. Therefore, while both schemes have a pre-processing step of a similar flavor, what is achieved by these steps is quite different. The representation of vqSGD is most efficient when it concentrates the information in a few coefficients, while Kashin's representation spreads the information evenly across different coefficients. The first representation serves us well when we only seek to quantize the signal. However, the quantized signal becomes very sensitive to privatization noise. Therefore vqSGD ends up with $O(d^2)$ error in the case of both privacy and communication constraints, while we can achieve $O(d)$ error.

Finally, we point out that SQKR easily extends to an optimal scheme for statistical mean estimation, where each local data is drawn from an unknown distribution $P$ supported on $\boldsymbol{B}_d(\boldsymbol{0}, 1)$, and the goal is to estimate the statistical mean. Under the statistical setting, however, SQKR requires no shared randomness. See Section D.3 for more details.

In Figure 1, we compare SQKR with a concatenation of separately optimal schemes [17] and [24], showing that under the same privacy and communication constraints, SQKR achieves much smaller estimation errors. More detailed experiments can be found in Section C.

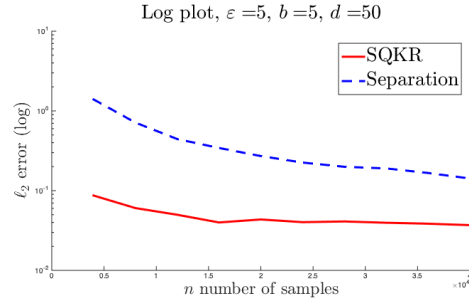

Figure 1: Log-scale $\ell_2$ error of mean estimation schemes.

## 3 Frequency Estimation

Recall that in the frequency estimation problem, given $X_1, ... X_n \in [d]$, we want to estimate the empirical frequency $D_{X^n}(x)$ under $\varepsilon$-LDP and $b$ bits communication budgets on each $X_i$. The following theorem characterizes the optimal estimation error achievable in this setting.

**Theorem 3.1** *For frequency estimation under $\varepsilon$-LDP and $b$ bits communication constraint, we can achieve*

*(i)* $r_{\mathsf{FE}}\left(\ell_2\right) \preceq \frac{d}{n\min\left\{e^\varepsilon, (e^\varepsilon - 1)^2, 2^b, d\right\}}$, *and* $r_{\mathsf{FE}}\left(\ell_1\right) \preceq \frac{d}{\sqrt{n\min\left\{e^\varepsilon, (e^\varepsilon - 1)^2, 2^b, d\right\}}}$;

*(ii)* $r_{\mathsf{FE}}\left(\ell_\infty\right) \preceq \sqrt{\frac{\log d}{n\min\left\{\varepsilon^2, \varepsilon, b\right\}}}$.

*Moreover, if* $\min\left(e^\varepsilon, (e^\varepsilon - 1)^2, 2^b\right) = o(d)$ *and* $n\min\left(e^\varepsilon, (e^\varepsilon - 1)^2, 2^b\right) \geq d^2$, *the errors in (i) are order-optimal.*

Note that, similar to Theorem 2.1, Theorem 3.1 shows that for finite $\varepsilon$ and $b$, the error is determined by the error due to one of these constraints, and hence the other less stringent constraint is satisfied

for free. It also implies that to achieve the optimal accuracy under $\varepsilon$-LDP constraints, we do not need more than $\min\left(\lceil \log_2 e \cdot \varepsilon \rceil, \log d\right)$ bits. In the rest of the section, we overview the scheme we develop to achieve the optimal error in (2).

We next overview the scheme that achieves the error in (i) of Theorem 3.1. We call this scheme Recursive Hadamard Response (RHR) as it builds on the recursive structure of the Hadamard matrix. The formal description of the scheme and complete proof of Theorem 3.1 can be found in Section E.

## 3.1 Recursive Hadamard Response

For notational convenience, we will view $D_{X^n}$ as a $d$-dimensional vector $(D_{X^n}(1), ..., D_{X^n}(d))$ and assume $X_i$ is one-hot encoded, i.e. $X_i = \boldsymbol{e}_j$ for some $j \in [d]$, so $D_{X^n} = \frac{1}{n}\sum_i X_i$. We further assume, without of loss of generality, that $d = 2^m$ for some $m \in \mathbb{N}$. Recall that a Hadamard matrix $H_d \in \{-1, +1\}^{d \times d}$ can be constructed in a recursive fashion as

$$H_m = \begin{bmatrix} H_{m/2} & H_{m/2} \\ H_{m/2} & -H_{m/2} \end{bmatrix},$$

where $H_1 = [1]$. It can be easily shown that $H_d^{-1} = H_d/d$.

Instead of directly estimating $D_{X^n}$, our strategy is to first estimate $H_d \cdot D_{X^n}$ and then perform the inverse transform $H_d^{-1}$ to get an estimate for $D_{X^n}$. So each client will transmit information about $Y_i \triangleq H_d \cdot X_i \in \{-1, 1\}^d$ rather than its original data $X_i$.

**The 1-bit case** In this case, each client transmits a uniformly at random chosen entry of $Y_i$ via any 1-bit LDP channel (for instance, using the 2-randomized response (RR) scheme [30, 32, 49]). Once receiving all the bits of the clients, the server can construct an unbiased estimator of $Y_i$ (since the randomness is public the server knows which entry is chosen for communication by each client). It turns out that this simple 1-bit scheme achieves optimal $\ell_1$ (and $\ell_2$) error $\Theta(\sqrt{d^2/n\varepsilon^2})$ in the high privacy regime $\varepsilon < 1$. This idea is not new and has been used in heavy hitter estimation [11] and distribution estimation [3]. However, a key question remains: *how do we minimize the error given an arbitrary communication budget $b$ and privacy level $\varepsilon$?*

**Moving beyond the 1-bit case** A natural way to extend the 1-bit scheme above to the case when each client can transmit $b$-bits is to have each client communicate $b$ randomly chosen entries of its transformed data $Y_i$ instead of a single entry. This will boost the sample size by a factor of $b$, equivalently decrease the $\ell_2$ error by a factor of $b$ ($\sqrt{b}$ for $\ell_1$). Instead, we argue next that we can exploit the recursive structure of the Hadamard matrix to boost the sample size by a factor of $2^b$, equivalently decrease the error by an exponential factor.

Consider $b \leq \lfloor \log d \rfloor$ and let $B = d/2^{b-1}$. Note that $H_d = H_{2^{b-1}} \otimes H_B$, where $\otimes$ denotes the Kronecker product. To visualize, for $b = 3$, $H_d$ has the following structure:

$$Y_i = H_d X_i = \begin{bmatrix} H_B & H_B & H_B & H_B \\ H_B & -H_B & H_B & -H_B \\ H_B & H_B & -H_B & -H_B \\ H_B & -H_B & -H_B & H_B \end{bmatrix} \begin{bmatrix} X_i^{(1)} \\ X_i^{(2)} \\ X_i^{(3)} \\ X_i^{(4)} \end{bmatrix},$$

where for $l = 1, \ldots, 2^{b-1}$, $X_i^{(l)}$ denotes the $l$'th block of $X_i$ of length $B = d/2^{b-1}$. Therefore, in order to communicate $Y_i$, we can equivalently communicate $H_B X_i^{(l)}$ for $l = 1, \ldots, 2^{b-1}$. Since $H_{2^{b-1}}$ is known, this is sufficient to reconstruct $Y_i$. We next observe that while communicating $Y_i$ requires $d = B \times 2^{b-1}$ bits, communicating $\{H_B X_i^{(l)}, l = 1, \ldots, 2^{b-1}\}$ requires $B + (b-1)$ bits. This is because $X_i$ is one-hot encoded and all but one of the $2^{b-1}$ vectors $\{H_B X_i^{(l)}, l = 1, \ldots, 2^{b-1}\}$ are equal to zero. It suffices to communicate the index $l$ of the non-zero vector, by using $(b-1)$ bits, and its $B$ entries by using additional $B$ bits. This is the key observation that RHR builds on.

When each client has only $b$ bits, they cannot communicate sufficient information for fully reconstructing $Y_i$, i.e. all $\{H_B X_i^{(l)}, l = 1, \ldots, 2^{b-1}\}$. Instead, each client chooses a random index $r_i \in [B]$ and communicates the $r_i$'th row of $\{H_B X_i^{(l)}, l = 1, \ldots, 2^{b-1}\}$, equivalently

$\{(H_B)_{r_i} X_i^{(l)}, l = 1, \ldots, 2^{b-1}\}$ where $(H_B)_{r_i}$ denotes the $r_i$'th row of $H_B$. Note that as before, only one of the $2^{b-1}$ numbers $\{(H_B)_{r_i} X_i^{(l)}, l = 1, \ldots, 2^{b-1}\}$ is non-zero and therefore these numbers can be communicated by using $b$ bits, $b-1$ bits to represent the index of the non-zero number and a single bit to communicate its value. When there is a privacy constraint, client $i$ perturbs their $b$ bits by a $2^b$-RR mechanism with privacy level $\varepsilon$, and this yields the privatized report of $b$ bits.

Upon receiving the reports from clients, the server constructs an unbiased estimator for $Y_i$. To do this, it first constructs an unbiased estimator for $\{H_B X_i^{(l)}, l = 1, \ldots, 2^{b-1}\}$ and then employs the structure $H_d = H_{2^{b-1}} \otimes H_B$. Note that since the randomness is shared the server knows the index $r$ chosen by each client, and since the clients choose their indices independently and uniformly at random, roughly speaking, they communicate information about different rows of $\{H_B X_i^{(l)}, l = 1, \ldots, 2^{b-1}\}$. Finally, an unbiased estimator $\hat{Y}_i$ for $Y_i$ yields an unbiased estimator for $X_i$ through the transformation $\hat{X}_i = \frac{1}{d} H_d \cdot \hat{Y}_i$, and due to the orthogonality of $H_d$, it can be shown that the variance of $\hat{X}_i$ is the same as the variance of $\hat{Y}_i$ divided by $d$.

A subtle issue is that if $e^\varepsilon \ll 2^b$, the noise due to $2^b$-RR mechanism may be too large, so instead of using all $b$ bits, we perform the above encoding and decoding procedure with $b' \triangleq \min \left( \lceil \log_2 e \cdot \varepsilon \rceil \right)$. We defer the details and the formal proof to Section E.1.

Note that this careful construction based on the recursive structure of the Hadamard matrix is only required in the case when there are joint privacy and communication constraints. When only one constraint is present, the optimal error can be achieved in a much simpler fashion. When there is only a $b$ bit constraint, [27] shows that the optimal error can be achieved by simply having each client communicate a subset of the entries of its data vector $X_i$ (without requiring Hadamard transform). When there is only a privacy constraint $\varepsilon$, the optimal error can be achieved by a number of schemes, such as subset selection ($2^b$-SS) [51] and Hadamard response (HR) [4].

**Remark 3.1** *As in mean estimation, RHR requires public randomness to achieve optimal communication efficiency. Indeed, we can show that RHR uses the minimum amount of shared randomness. See Section B for more details.*

## 3.2 Application to distribution estimation

For frequency estimation, RHR requires shared randomness so that the server can construct an unbiased estimator. However, for distribution estimation where $X_i \sim \boldsymbol{p}$, we can replace the random sampling with a deterministic partitioning of coordinates among the different clients and circumvent the need for shared randomness. This gives us the following theorem:

**Theorem 3.2** *For distribution estimation under $\varepsilon$-LDP and $b$-bit communication constraints,*

$$r_{\text{DE}}(\ell_2) \asymp \frac{d}{n \min \left( e^\varepsilon, (e^\varepsilon - 1)^2, 2^b, d \right)}, \text{ and } r_{\text{DE}}(\ell_1) \asymp \frac{d}{\sqrt{n \min \left( e^\varepsilon, (e^\varepsilon - 1)^2, 2^b, d \right)}},$$

*without shared randomness. Moreover, if $n \cdot \min \left( e^\varepsilon, (e^\varepsilon - 1)^2, 2^b, d \right) \geq d^2$, the above errors are optimal even in the presence of shared randomness.*

The lower bounds follow directly from the results of [51] (under LDP constraint) and [9, 27] (under communication constraint). We leave the formal proof of the achievability to Section F.

We compare RHR with other distribution estimation mechanisms, including state-of-the-art methods HR [4], 1-bit HR [3] and SS [51]. We generate data from different distributions and compare the $\ell_1$ errors of each schemes. In Figure 2, we set $d = 10000$, and consider truncated and normalized geometric distribution with $\lambda = 0.8$. The figure shows that our scheme achieves better performance than [4] with half the communication cost. More experiments can be found in Section C.

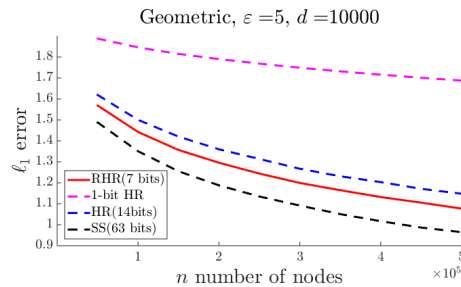

Figure 2: $\ell_1$ estimation errors of different distribution estimation schemes.

## 4    Acknowledgments

This work was supported in part by a Google Faculty Research Award and by the National Science Foundation under Grant CCF-1704624. The authors would like to thank Jakub Konečný for bringing Kashin's representation to their attention. This was helpful in achieving order-optimality for mean estimation.

## Broader Impact

Harnessing distributed data holds the promise of impacting many facets of our lives. It could enable truly large scale smart infrastructure and IoT applications; having a profound and positive impact on power-grid efficiency, traffic, health-monitoring, medical diagnoses, carbon emissions, and many other areas. A foundational understanding of distributed learning and estimation can also benefit many different fields of study such as neuroscience, medicine, economics, and social networks, where statistical tools are often used to analyze information that is generated and processed in large networks.

While the above vision is expected to generate many disruptive business and social opportunities, it presents a number of unprecedented challenges. First, massive amounts of data need to be collected by, and transferred across, resource-constrained devices. Second, the collected data needs to be stored, processed, and analyzed at scales never previously seen. Third, serious concerns such as access control, data privacy, and security should be rigorously addressed.

Our work tackles the above challenges by examining the fundamental trade-off between communication, privacy, and accuracy by taking a holistic approach that examines all these constraints simultaneously and designing provably optimal privacy and compression mechanisms for efficient distributed learning and estimation. Our work therefore serves as a fundamental stepping stone towards harnessing large-scale distributed data in a privacy-preserving and bandwidth efficient way.

Even more broadly, our work advances the current state-of-the-art in a number of areas of statistics and computer science. Indeed, our work builds on a long line of fundamental research in information theory, statistics, and theoretical computer science, extending them in non-trivial ways.

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
