[Supplementary Material]

## A  Separate Quantization and Privatization Is Strictly Sub-optimal

**Distribution estimation**  First let us recap the subset selection (SS) scheme proposed by [51]. Assume $X_1, ..., X_n \overset{\text{i.i.d.}}{\sim} \boldsymbol{p} = (p_1, ..., p_d)$. Client $i$ maps the local data $X_i$ into $y \in \mathcal{Y}_{d,w} \triangleq \left\{ y \in \{0,1\}^d : \sum_j y_j = w \right\}$ with the transitional probability

$$Q_{\text{SS}}(y|X = j) = \frac{e^\varepsilon y_j + (1 - y_j)}{e^\varepsilon \binom{d-1}{w-1} + \binom{d-1}{w}}.$$

The estimator for $p_j$ is defined by

$$\hat{p}_j \triangleq \left( \frac{(d-1)e^\varepsilon + \frac{(d-1)(d-w)}{w}}{(d-w)(e^\varepsilon - 1)} \right) \frac{T_j}{n} - \frac{(w-1)e^\varepsilon + d - w}{(d-w)e^\varepsilon - 1}, \tag{3}$$

where $T_j \triangleq \sum_{i=1}^n Y_i(j)$. Note that by picking $w = \lceil \frac{d}{e^\varepsilon + 1} \rceil$, SS is order-optimal for all privacy regimes.

To demonstrate that separating privatization and quantization is strictly sub-optimal, we analyze the estimation error of directly concatenating the $2^b$-SS mechanism with the grouping-based quantization in [27]. Note that both schemes are known to be optimal under the corresponding constraints, privacy and communication respectively. However, their direct combination yields an $\ell_2$ error of order $O\left(d^2\right)$, which is far from the optimal accuracy established in Theorem 3.1.

We first group $[d]$ into $s = d/2^b$ equal-sized groups $\mathcal{G}_1, ..., \mathcal{G}_s$, and each client is only responsible to send information about one particular group. That is, let $Y_i$ be the outcome of the $2^b$-SS mechanism, i.e. $Y_i \sim Q_{\text{SS}}(\cdot|X_i)$, and client $i$ only transmits $\{Y_i(j)|j \in \mathcal{G}_{s'}\}$, for some $s' \in [s]$. Since the server estimates each component of $\boldsymbol{p}$ separately as in (3), this grouping strategy reduces the effective sample size from $n$ to $n' = n2^b/d$. Plugging $n'$ into the $\ell_2$ error (see Proposition III.1 in [51]), we conclude that the error grows as

$$O\left( \frac{d^2}{n2^b \min\left(e^\epsilon, (e^\epsilon - 1)^2\right)} \right).$$

Note that since each $Y_i$ contains exactly $w$ ones, the required communication budget to describe $\{Y_i(j), j \in \mathcal{G}_l\}$ may be larger than $b$ bits. But this is fine since it implies that even given more than $b$ bits, the estimation error still grows with $d^2$. In Theorem 3.2, on the other hand, we show that the optimal $\ell_2$ error is linear in $d$, so this demonstrates that separate quantization and privatization is sub-optimal.

**Mean estimation**  For the mean estimation problem, a straightforward combination is using the `PrivUnit` mechanism (see Algorithm 1 in [13]) to perturb the local data $X_i \in \mathcal{B}_d(\mathbf{0}, 1)$, and then using `RandomSampling` quantization in (Theorem 6 in [24]) to compress the perturbed data. Both schemes are known to be optimal under the corresponding constraints, privacy and communication respectively. (Note that in the implementation, we replaced the `RandomSampling` quantization with a Kashin's quantizer, since implementing the theoretically optimal `RandomSampling` quantizaton is computationally infeasible.)

By Proposition 4 in [13], the output of `PrivUnit`, denoted as $Z_i = \text{PrivUnit}(X_i, \varepsilon)$, has $\ell_2$ norm of order $\Theta\left(\sqrt{\frac{d}{\min(\varepsilon, \varepsilon^2)}}\right)$. However, if we further apply `RandomSampling` to $b$ bits, by Theorem 6 in [24], the $\ell_2$ estimation error grows as

$$\Theta\left( \|Z_i\| \frac{d}{n \cdot b} \right) = \Theta\left( \frac{d^2}{nb \min\left(\varepsilon, \varepsilon^2\right)} \right),$$

showing a quadratic dependence in $d$. By Theorem 2.1, nevertheless, we can construct a better scheme with $O(d/n \min\left(\varepsilon, \varepsilon^2, b\right))$ dependence under both constraints.

# B   Role of Shared Randomness and How It Benefits Communication

**The Amount of Shared Randomness**   In the achievability part of Theorem 2.1, our proposed scheme SQKR randomly and independently samples $b_{\text{ME}}^* \triangleq \min\left(\lceil \varepsilon \rceil, b\right)$ bits from the quantized $d$-dimensional binary vector at each client. These bits are then privatized and communicated to the server. In addition to the values of these bits, the server needs to know the indices of the sampled bits, which corresponds to an additional $b_{\text{ME}}^* \log d$ bits of information that needs to be shared between each client and the server. This information can be shared in two different ways: 1) sampling can be done by using a public coin shared a priori between the client and the server, or 2) sampling can be done by using a private coin at the client side, which is then communicated to the server. We can also combine both 1) and 2) when $b > b_{\text{ME}}^*$: given $b$ bits communication budget, SQKR compresses the data to $b_{\text{ME}}^*$ bits, so the client can use the remaining $b - b_{\text{ME}}^*$ bits to communicate the locally generated randomness required at the sampling step. Thus the amount of shared randomness is reduced to $b_{\text{ME}}^* \log d - (b - b_{\text{Me}}^*)$ bits. Moreover, by extending [3, Theorem 4], we also obtain a lower bound on the amount of shared randomness required, which we summarize in the following corollary:

**Corollary B.1**   *Under $\varepsilon$-LDP and $b$-bit communication constraints, SQKR uses $\min\left(b_{\text{ME}}^* \log d, d\right) - (b - b_{\text{ME}}^*)$ bits of shared randomness to achieve $r_{\text{ME}}\left(\ell_2, b, \varepsilon\right)$, where $b_{\text{ME}}^* \triangleq \min\left(\lceil \varepsilon \rceil, b\right)$. Moreover, if $b < \log d - 2$, any $b$-bit consistent mean estimation scheme[1] requires at least $\log d - b - 2$ bits.*

We contrast this with the amount of shared randomness needed in the generic scheme of [12] which provides $\varepsilon$-LDP by using 1 bit per client in the high privacy regime $\varepsilon = O(1)$. The shared randomness required by this scheme is $d$ bits per client. In contrast, when $\varepsilon = O(1)$ and $b = 1$, SQKR requires $\log d$ bits of shared randomness.

Similarly, for frequency estimation, it can be seen that RHR requires $\log d - b_{\text{FE}}^*$ bits of shared randomness in the random sampling step, where $b_{\text{FE}}^* \triangleq \min\left(\lceil \varepsilon \log_2 e \rceil, b\right)$. Again, this is achieved by communicating $b - b_{\text{FE}}^*$ bits of privately generated randomness from the client to the the server, which reduces the required public randomness to $\log d - b$ bits. Furthermore, as in mean estimation, we can show that at least $\log d - b - 2$ bits are needed to get a consistent scheme, so RHR is also optimal in the amount of public randomness it uses. We summarize it in the following corollary:

**Corollary B.2**   *Under $\varepsilon$-LDP and $b$-bit communication constraints, RHR uses $\log d - b$ bits of shared randomness to achieve $r_{\text{FE}}\left(\ell_2, b, \varepsilon\right)$, where $b_{\text{FE}}^* \triangleq \min\left(\lceil \varepsilon \log_2 e \rceil, b\right)$. Moreover, if $b < \log d - 2$, any $b$-bit consistent frequency estimation scheme requires at least $\log d - b - 2$ bits of shared randomness. Thus RHR is optimal in the amount of shared randomness it uses for frequency estimation, up to an additive constant.*

The achievability parts of Corollary B.1 and Corollary B.2 follow directly from the analysis of SQKR and RHR, and we defer the proof of the converse part to Section G.2. Given a $\varepsilon$-LDP constraint, we summarize the minimum amounts of communication and shared randomness required to achieve the optimal error $r_{\text{ME}}\left(\ell_2, \varepsilon, \infty\right)$ and $r_{\text{FE}}\left(\ell_2, \varepsilon, \infty\right)$ in Table 4.

|  | Communication | Shared randomness |
|---|---|---|
| SQKR (Thm. 2.1) | $\lceil \varepsilon \rceil$ bits | $\min\left(\lceil \varepsilon \rceil \log d, d\right)$ bits |
| RHR (Thm. 3.1) | $\lceil \log_2 e \cdot \varepsilon \rceil$ bits | $\log d - \lfloor \log_2 e \cdot \varepsilon \rfloor$ bits |

Table 4: The amounts of required shared randomness.

In Figure 3, we plot the achievable region for the minimax frequency estimation error under $\varepsilon$-LDP constraint (i.e. $r_{\text{FE}}\left(\ell_2, \varepsilon, \infty\right)$). Note that the red line in Figure 3 can be achieved by RHR.

**Remark B.1**   *Note that shared randomness is only needed for distribution-free settings; for distribution estimation and statistical mean estimation, one can achieve the same estimation error with only private randomness as noted in Theorems D.1 and 3.2.*

Figure 3: Achievable region for frequency estimation with public randomness.

**Converting public-coin schemes to private-coin schemes** As discussed above, we can always replace shared randomness with additional communication by first generating the random bits at the client side and then sending them to the server. Therefore, by Corollary B.1 and Corollary B.2, we automatically obtain private-coin SQKR and private-coin RHR by using additional communication. We next state these observations for completeness.

**Corollary B.3 (Private-coin SQKR)** *Under $\varepsilon$-LDP and $b$-bit communication constraints with $b > \log d$ and $0 < \varepsilon \leq d$, the $\ell_2$ minimax error for private-coin mean estimation, denoted as $\tilde{r}_{ME}(\ell_2, \varepsilon, b)$[2] (to distinguish it from the minimax error $r_{ME}(\ell_2, \varepsilon, b)$ achieved by public-coin schemes), is characterized as follows:*

*(i) if $\log d < b < d$, then*

$$\tilde{r}_{ME}(\ell_2, \varepsilon, b) \preceq \frac{d}{n \min\left(\varepsilon^2, \varepsilon, b/\log d, d\right)};$$

*(ii) if $b \geq d$, then*

$$\tilde{r}_{ME}(\ell_2, \varepsilon, b) \preceq \frac{d}{n \min\left(\varepsilon^2, \varepsilon, d\right)},$$

*and the above errors can be achieved by private-coin SQKR. Therefore private-coin SQKR requires $O\left(\min\left(\lceil \varepsilon \rceil \log d, d\right)\right)$ bits of communication to achieve $\tilde{r}_{ME}(\ell_2, \varepsilon, \infty)$.*

Similarly, the estimation error of private-coin RHR is characterized below:

**Corollary B.4 (Private-coin RHR)** *Under $\varepsilon$-LDP and $b$-bit communication constraints with $b > \log d$ and $0 < \varepsilon \leq \log d$, the $\ell_2$ minimax error for private-coin frequency estimation, denoted as $\tilde{r}_{FE}(\ell_2, \varepsilon, b)$, is*

$$\tilde{r}_{FE}(\ell_2, \varepsilon, b) \preceq \frac{d}{n \min\left((e^\varepsilon - 1)^2, e^\varepsilon, d\right)},$$

*which can be achieved by private-coin RHR. In words, for any $\varepsilon$, private-coin RHR always uses $\log d$ bits of communication to achieve $\tilde{r}_{FE}(\ell_2, \varepsilon, \infty)$.*

Moreover, the following lemma, an extension of [3, Theorem 4], establishes a lower bound on the communication required for consistent private-coin schemes:

**Lemma B.1** *Any consistent private-coin scheme for both mean estimation and frequency estimation uses at least $b > \log d - 2$ bits of communication.*

This shows that the $\log d$ lower bounds on $b$ in both corollaries are fundamental (within 2 bits). The proof of the lemma is given in Section G.

# C  Experiments

In this section, we implement our mean estimation and frequency estimation schemes and present our experimental results[3]. More detailed results can be found in Section C.

## C.1  Mean estimation

We implement our mean estimation scheme Subsampled and Quantized Kashin's Response (SQKR) as in Section 2 and compare it with 1) an optimal $\varepsilon$-LDP mechanism `privUnit` [13], and 2) a baseline under both LDP and communication constraints – a concatenation of `privUnit` [13] (which is order-optimal under $\varepsilon$-LDP) and the quantizer based on Kashin's representation [36] (which is optimal up to a logarithmic factor, under $b$-bit communication constraint).

**Generating the data**    In order to capture the distribution-free setting, we generate data independently but non-identically; in particular, we set $Z_1, ..., Z_{n/2} \overset{i.i.d.}{\sim} N(1,1)^{\otimes d}$ and $Z_{n/2+1}, ..., Z_n \overset{i.i.d.}{\sim} N(10,1)^{\otimes d}$ (this also makes the data non-central, i.e. $\mathbb{E}\left[\sum Z_i\right] \neq 0$). Since each sample has bounded $\ell_2$ norm, we normalize each $Z_i$ by setting $X_i = Z_i / \|Z_i\|_2$.

**Generating the tight frame**    We construct the tight frame by using the random partial Fourier matrices in [36]. Specifically, we set $N = 2^{\lceil \log_2 d \rceil + 1} = \Theta(d)$, and choose the basis $U = \left\{1/\sqrt{N}, -1/\sqrt{N}\right\}^{N \times d}$ by selecting the first $d$ rows of $H_N \cdot D$, where $H_N$ is a $N \times N$ Hadamard matrix and $D$ is a random diagonal matrix with each diagonal entry generated from uniform $\{+1, -1\}$. It can be shown that the tight frame based on $U$ has Kashin's level $K = \tilde{O}(1)$.

**Compare to optimal $\varepsilon$-LDP scheme [13]**

Figure 4: $\ell_2$ error of `privUnit` and SQKR with different dimensions $d = 50, 200$.

We first compare our scheme SQKR with `privUnit` [13], which is order-optimal under $\varepsilon$-LDP. Since the outcome of `privUnit` is a $d$-dimensional vector lying in a radius $O(\sqrt{d})$ sphere, in general we need $32d$ bits to represent it (where we assume each float requires 32 bits). Figure 4 shows that SQKR achieves similar performance with significantly communication budgets. For instance, when $\varepsilon = 5$ and $d = 50$, the communication cost of `privUnit` is $2K$ bits, while SQKR uses only 5 bits but attains similar performance.

**Compare with the baseline scheme**

Next, we compare SQKR with a combination of `privUnit` and an optimal quantizer.

**Baseline: a direct concatenation of `privUnit`, Kashin's quantizer and sampling** For each $X_i$ in unit $\ell_2$ ball, `privUnit` maps it to a vector $\tilde{X}_i$ with length $\left\| \tilde{X}_i \right\|_2 = \Theta \left( \sqrt{d / \min\left( \varepsilon, \varepsilon^2 \right)} \right)$. If we quantize $\tilde{X}_i$ according to its Kashin's representation and then subsample $b$ bits from it as in Section 2, then the $\ell_2$ error (i.e. variance) will be

$$\tilde{O} \left( \frac{d}{b} \left\| \tilde{X}_i \right\|^2 \right) = \tilde{O} \left( \frac{d^2}{b \min\left( \varepsilon, \varepsilon^2 \right)} \right).$$

Therefore, averaging over $n$ clients, the $\ell_2$ error of estimating the empirical mean is

$$\tilde{O} \left( \frac{d^2}{n \cdot b \min\left( \varepsilon, \varepsilon^2 \right)} \right).$$

However, in Theorem 2.1, we see that with a more sophisticated design, we can achieve smaller $\ell_2$ error

$$O \left( \frac{d}{n \cdot \min\left( \varepsilon, \varepsilon^2, b \right)} \right).$$

**Setup** In the experiment, we mainly focus on the *high-privacy low-communication* setting where $\varepsilon = b = 1$, and the *low-privacy high-communication* setting where $\varepsilon = b = 5$. We consider different dimensions $d$ and plot the (log-scale) $\ell_2$ estimation error (i.e. mean square error) with sample size $n$. For each point, i.e. each combination of parameters $\varepsilon, b, d, n$, we repeat the simulation for 8 iterations and compute the average. In Figure 5, we see that SQKR drastically outperforms the baseline (labeled as "Separation" since it is based on the idea of separately coding for privacy and communication efficiency). The gain increases in higher dimensions or with more stringent privacy/communication constraints.

Figure 5: Log-scale $\ell_2$ error with different dimensions $d = 20, 50, 80$ and different privacy and communication budgets.

In order to study the dependence on $d$, we fix the sample size to $n = 10^5$ and $\varepsilon, b$, and increase the dimension $d$. In Figure 6, We see that SQKR has linear dependence on $d$, and `Separation` has super-linear dependence. Therefore the performance differs drastically when $d$ increases.

Figure 6: $\ell_2$ error with $n = 10^5$ and different dimensions $d$. In order to better emphasize the dependence to $d$, on the right-hand side we only plot the $\ell_2$ error of SQKR.

## C.2 Frequency estimation

For frequency estimation, we compare our scheme, Recursive Hadamard Response (RHR), with SS [51], HR [4] and 1-bit HR [3][4]. We set $d = \{1000, 5000, 10000\}$, $\varepsilon \in \{0.5, 2, 5\}$ and $n = \{50000, 100000, ..., 500000\}$, and evaluate the $\ell_1$ estimation errors on uniform distribution and truncated and normalized geometric distribution with $\lambda = 0.8$. For each point (i.e. for each parameter $n, \varepsilon, d$), we repeat the simulation 30 times and average the $\ell_2$ errors. Figure 7 and Figure 8 show that RHR can achieve the same performance as HR but is significantly more communication efficient. For instance, in Figure 8 with $d = 10000, \varepsilon = 5$, RHR uses only half of the communication budget for HR and achieves better performance. In all settings, $k$-SS has the best statistical performance, but this comes with drastically higher communication and computation cost.

Figure 7: $\ell_1$ error with $d = 1000$. Left are $Geo(0.8)$ and right are *Uniform*.

Figure 8: $\ell_1$ error with $d = 5000$ and $d = 10000$, under (truncated) $Geo(0.8)$ and different $\varepsilon$.

In Figure 9, we record the decoding time for each scheme. The decoding complexity of RHR is similar to HR and 1-bit HR, which are all much more computationally efficient than SS.

Figure 9: Left: time complexity with $d = 1000, \varepsilon = 7$ right: time complexity with $d = 5000, \varepsilon = 2$.

# D Proof of Theorem 2.1

## D.1 Achievability

In this section, we prove that Subsampled and Quantized Kashin's Response (SQKR) achieves optimal $\ell_2$ estimation error. For each observation $X_i$, we will construct an unbiased estimator $\hat{X}_i$ (i.e. $\mathbb{E}\left[\hat{X}_i | X_i\right] = X_i$), where $\hat{X}_i$ is $\varepsilon$-LDP, can be described by $k$ bits, and has small variance. The encoding scheme consists of three main steps: (1) obtaining a Kashin's representation for a tight frame [36], (2) subsampling and (3) privatization.

**Kashin's representation** We begin with introducing tight frames and Kashin's representation [36].

**Definition D.1 (Tight frame)** *A tight frame is a set of vectors $\{u_j\}_{j=1}^N \in \mathbb{R}^d$ that obeys Parseval's identity*

$$\|x\|_2^2 = \sum_{j=1}^N \langle u_j, x\rangle^2, \text{ for all } x \in \mathbb{R}^d.$$

A frame can be viewed as a generalization of an orthogonal basis in $\mathbb{R}^d$, which can improve the encoding stability by adding redundancy to the representation system when $N > d$. To increase robustness, we wish the information to spread evenly in each coefficient, which motivates the following definition of a Kashin's representation:

**Definition D.2 ( Kashin's representation)** *For a set of vectors $\{u_j\}_{j=1}^N$, we say the expansion*

$$x = \sum_{j=1}^N a_j u_j, \text{ with } \max_j |a_j| \leq \frac{K}{\sqrt{N}} \|x\|_2$$

*is a Kashin's representation of vector $x$ at level $K$ .*

Therefore, if we can obtain unbiased estimators $\{\hat{a}_j\}_{j=1}^N$ of the Kashin's representation of $X$ with respect to a tight frame $\{u_j\}_{j=1}^N$, then the MSE can be controlled by

$$\mathbb{E}\left[\left(\hat{X} - X\right)^2\right] = \mathbb{E}\left[\left\|\sum_{j=1}^N (\hat{a}_j - a_j) u_j\right\|_2^2\right] \overset{(a)}{\leq} \mathbb{E}\left[\sum_{j=1}^N (\hat{a}_j - a_j)^2\right] = \sum_{j=1}^N \text{Var}\left(\hat{a}_j\right), \quad (4)$$

where (a) is due to the Cauchy–Schwarz inequality and the definition of a tight frame. Recall that $X$ is deterministic, so here the expectation is taken with respect to the randomness on $\hat{a}_j$. Notice that the cardinality $N$ of the frame determines the compression (i.e. quantization) rate, and Kashin's level $K$ affects the variance. Hence we are interested in constructing tight frames with small $N$ and $K$.

By Theorem 3.5 and Theorem 4.1 in [36], we have the following lemma:

**Lemma D.1 (Uncertainty principle and Kashin's Representation)** *For any $\mu > 0$ and $N > (1 + \mu)d$, there exists a tight frame $\{u_j\}_{j=1}^N$ with Kashin's level $K = O\left(\frac{1}{\mu^3}\log\frac{1}{\mu}\right)$. Moreover, for each $X$, finding Kashin's coefficient requires $O\left(dN\log N\right)$ computation.*

For our purpose, we choose $\mu$ to be a constant, i.e. $\mu = \Theta(1)$, so $N = \Theta(d)$, $K = \Theta(1)$, and we can obtain a representation of $X = \sum_{j=1}^N a_j u_j$, with $|a_j| \leq \frac{K}{\sqrt{N}} = \frac{c}{\sqrt{d}}$ for some constant $c$. Therefore, we quantize each $a_j$ as follows:

$$q_j \triangleq \begin{cases} -\frac{c}{\sqrt{d}}, & \text{with probability } \frac{c/\sqrt{d}-a_j}{2c/\sqrt{d}} \\ \frac{c}{\sqrt{d}}, & \text{with probability } \frac{a_j+c/\sqrt{d}}{2c/\sqrt{d}}. \end{cases} \quad (5)$$

$\boldsymbol{q} \triangleq (q_1, ..., q_N)$ yields an unbiased estimator of $\boldsymbol{a} \triangleq (a_1, ..., a_N)$ and can be described by $N = \Theta(d)$ bits.

**Sampling** To further reduce the communication cost, we sample $k$ bits uniformly at random from $\boldsymbol{q}$ using public randomness. Let $s_1, ..., s_k \overset{\text{i.i.d.}}{\sim} \text{uniform}[N]$ be the indices of the sampled elements, and define the sampled message as

$$Q\left(\boldsymbol{q}, (s_1, ..., s_k)\right) = (q_{s_1}, ..., q_{s_k}) \in \left\{-c/\sqrt{d}, c/\sqrt{d}\right\}^k. \tag{6}$$

Then $Q$ can be described in $k$ bits, and each of $q_{s_m}$ yields an independent and unbiased estimator of $\boldsymbol{a}$:

$$\mathbb{E}\left[N \cdot q_{s_m} \cdot \mathbb{1}_{\{j=s_m\}}\right] = \mathbb{E}\left[\mathbb{E}\left[N \cdot q_{s_m} \cdot \mathbb{1}_{\{j=s_m\}} \big| q_1, ..., q_N\right]\right] = \mathbb{E}\left[q_j\right] = a_j, \ \forall j \in [N]. \tag{7}$$

**Privatization** Each client then perturbs $Q$ via $2^k$-RR mechanism (as a $k$-bit string):

$$\tilde{Q} = \begin{cases} Q, & \text{with probability } \frac{e^\varepsilon}{e^\varepsilon + 2^k - 1} \\ Q' \in \left\{-c/\sqrt{d}, c/\sqrt{d}\right\}^k / \{Q\}, & \text{with probability } \frac{1}{e^\varepsilon + 2^k - 1}. \end{cases} \tag{8}$$

Since

$$\sum_{Q' \in \{-c/\sqrt{d}, c/\sqrt{d}\}^k / \{Q\}} Q' = -Q,$$

it is not hard to see $\left(\frac{e^\varepsilon + 2^k - 1}{e^\varepsilon - 1}\right) \tilde{Q}$ yields an unbiased estimator of $Q$. Indeed, if we write $\tilde{Q} = (\tilde{q}_1, ..., \tilde{q}_k)$, then

$$\mathbb{E}\left[\left(\frac{e^\varepsilon + 2^k - 1}{e^\varepsilon - 1}\right) \cdot \tilde{q}_m \bigg| q_1, ..., q_N, s_1, ..., s_k\right] = q_{s_m}, \tag{9}$$

or equivalently

$$\mathbb{E}\left[\left(\frac{e^\varepsilon + 2^k - 1}{e^\varepsilon - 1}\right) \tilde{Q} \bigg| Q\right] = Q.$$

**Estimation and the $\ell_2$ error** Given $\tilde{Q} = (\tilde{q}_1, ..., \tilde{q}_k)$, define

$$\hat{a}_j = \frac{N}{k} \cdot \left(\frac{e^\varepsilon + 2^k - 1}{e^\varepsilon - 1}\right) \sum_{m=1}^{k} \tilde{q}_m \cdot \mathbb{1}_{\{j=s_m\}}.$$

According to (7) and (9), $\mathbb{E}[\hat{a}_j] = a_j$, and hence $\hat{X}\left(\tilde{Q}, (s_1, ..., s_k)\right) \triangleq \sum_{j=1}^{N} \hat{a}_j u_j$ gives us an unbiased estimator of $X$.

**Claim D.1** *The MSE of $\hat{X}$ can be bounded by*

$$\mathbb{E}\left[\left\|\hat{X} - X\right\|_2^2\right] \leq C \left(\frac{e^\varepsilon + 2^k - 1}{e^\varepsilon - 1}\right)^2 \frac{d}{k}.$$

Finally, each client encodes its data $X_i$ independently, and the server computes $\frac{1}{n} \sum_i \hat{X}_i$. Since $\hat{X}_i$ is unbiased and by Claim D.1, we get

$$\mathbb{E}\left[\left\|\frac{1}{n} \sum_{j=1}^{n} \hat{X}_i - \bar{X}\right\|_2^2\right] = \frac{1}{n^2} \sum_{j=1}^{n} \mathbb{E}\left[\left\|\hat{X}_i - X_i\right\|_2^2\right] \leq C \left(\frac{e^\varepsilon + 2^k - 1}{e^\varepsilon - 1}\right)^2 \frac{d}{nk}.$$

Finally, picking $k = \min\left(\lceil \log_2 e \rceil \varepsilon, b\right)$ gives us the desired upper bound.

### D.2 Lower Bound of Theorem 2.1

As in the converse part of Theorem 3.1, the lower bound can be obtained by constructing a prior distribution on $X_i$ and analyzing the statistical mean estimation problem. Therefore, we will impose a prior distribution $P$ on $X_1, ..., X_n$ and lower bound the $\ell_2$ error of estimating the mean $\theta(P)$, where $P$ is a distribution supported on the $d$-dimension unit ball.

For any $\hat{X}$, observe that

$$\mathbb{E}_{\hat{X}, X^n \overset{\text{i.i.d.}}{\sim} P} \left[ \left\| \hat{X} - \bar{X} \right\|_2^2 \right] \overset{\text{(a)}}{\geq} \mathbb{E} \left[ \left( \left\| \hat{X} - \theta(P) \right\|_2 - \left\| \bar{X} - \theta(P) \right\|_2 \right)^2 \right]$$

$$\geq \mathbb{E} \left[ \left\| \hat{X} - \theta(P) \right\|_2^2 \right] - 2\mathbb{E} \left[ \left\| \hat{X} - \theta(P) \right\|_2 \left\| \bar{X} - \theta(P) \right\|_2 \right]$$

$$\overset{\text{(b)}}{\geq} \mathbb{E} \left[ \left\| \hat{X} - \theta(P) \right\|_2^2 \right] - 2\sqrt{\mathbb{E} \left[ \left\| \hat{X} - \theta(P) \right\|_2^2 \right] \mathbb{E} \left[ \left\| \bar{X} - \theta(P) \right\|_2^2 \right]},$$

(10)

where (a) and (b) follow from the triangular inequality and the Cauchy-Schwartz inequality respectively. Since $X_i$ and $\theta(P)$ are supported on the unit ball, $\mathbb{E} \left[ \left\| \bar{X} - \theta(P) \right\|_2^2 \right] \asymp 1/n$, so it remains to find a distribution $P^*$ such that

$$\min_{\hat{X}} \mathbb{E} \left[ \left\| \hat{X} - \theta(P^*) \right\|_2^2 \right] \succeq \frac{d}{n \min(\varepsilon^2, \varepsilon, b)}.$$

Consider the product Bernoulli model $Y \sim \prod_{j=1}^d \text{Ber}(\theta_j)$. If we set $\Theta = [1/2 - \varepsilon, 1/2 + \varepsilon]^d$ for some $\frac{1}{2} > \varepsilon > 0$, then it can be shown that both variance and sub-Gaussian norm of the score function of this model is $\Theta(1)$ [9, Corollary 4]. Therefore, applying [9, Corollary 8] and [8, Proposition 2, Proposition 4] yields

$$\min_{\hat{\theta}} \mathbb{E} \left[ \left\| \hat{\theta} - \theta \right\|_2^2 \right] \succeq \frac{d^2}{n \min(\varepsilon^2, \varepsilon, b)}.$$

Finally, if we set $X_i = Y_i/\sqrt{d}$, then each $X_i$ is supported on the unit ball and $\mathbb{E}[X_i] = \theta/\sqrt{d}$. Therefore

$$\min_{\hat{X}} \mathbb{E} \left[ \left\| \hat{X} - \frac{\theta}{\sqrt{d}} \right\|_2^2 \right] \succeq \frac{d}{n \min(\varepsilon^2, \varepsilon, b)}.$$

Plugging into (10), as long as $\min(\varepsilon^2, \varepsilon, k) = o(d)$, the first term dominates and we get the desired lower bound. $\square$

## D.3 Application to statistical mean estimation

For mean estimation, SQKR requires shared randomness so that the server can construct an unbiased estimator. However, for distribution estimation where $X_1, ..., X_n \overset{\text{i.i.d.}}{\sim} P$, we can replace the random sampling with a deterministic partitioning of coordinates among the different clients and circumvent the need for shared randomness. This gives us the following theorem:

**Theorem D.1** *For statistical mean estimation under $\varepsilon$-LDP and $b$ bits communication constraint, we can achieve*

$$r_{\text{SME}}(\ell_2, \varepsilon, b) \preceq \frac{d}{n \min(\varepsilon^2, \varepsilon, b, d)},$$

(11)

*without shared randomness. Moreover, if $\min(\varepsilon^2, \varepsilon, b) = o(d)$, the above error is optimal (even in the presence of shared randomness).*

**Proof.**

The lower bounds follow directly from [13] (under $\varepsilon$-LDP constraint) and [42] (under $b$-bit communication constraint). For the achievability part, we apply SQKR except that replacing the random sampling step by deterministic grouping.

Let $X_i \overset{\text{i.i.d.}}{\sim} P$ with $P$ supported on $\mathcal{B}(\mathbf{0}, 1)$. First, as in the proof of Theorem 3.1, by Lemma D.1 we can write $X_i = \sum_{j=1}^N A_{ij} u_j$ with $N = c_0 d$ and $|A_{ij}| \leq K/\sqrt{d}, K = \Theta(1)$. Since $X_i \overset{\text{i.i.d.}}{\sim} P$, if we denote $A_i = [A_{i1}, ..., A_{iN}]$, then $A_i \overset{\text{i.i.d.}}{\sim} Q$ for some $Q$ supported on $\left[ -\frac{K}{\sqrt{d}}, \frac{K}{\sqrt{d}} \right]^N$.

Now we group $n$ clients into $m \triangleq N/b^*$ groups $\mathcal{G}_1, ..., \mathcal{G}_m$, each with $nb^*/N$ clients, where $b^* \triangleq \min(\lceil \varepsilon \log_2 e \rceil, b)$. Also, we divide all of $N$ coordinates (of $A_i$) into $m$ groups $\mathcal{I}_1, ..., \mathcal{I}_m$, and

each group of clients are responsible for estimating the corresponding group of coordinates of $\theta\left(Q\right) \in \left[-\frac{K}{\sqrt{d}}, \frac{K}{\sqrt{d}}\right]^{N}$, where $\theta\left(Q\right) = \mathbb{E}_{Q}[A]$ is the mean of $Q$ and $\theta\left(Q\right)$.

**Quantization**   If client $i$ belongs to $\mathcal{G}_{l}$, then it quantizes $A_{ij}$ to $Q_{ij}$ according to

$$Q_{ij} \triangleq \begin{cases} -\frac{K}{\sqrt{d}}, & \text{with probability } \frac{K/\sqrt{d}-A_{ij}}{2K/\sqrt{d}}, \text{ if } j \in \mathcal{I}_{l}, \\ \frac{K}{\sqrt{d}}, & \text{with probability } \frac{A_{ij}+K/\sqrt{d}}{2K/\sqrt{d}}, \text{ if } j \in \mathcal{I}_{l}, \\ 0, & \text{else.} \end{cases} \tag{12}$$

Conditioned on $A_{i}$, $\{Q_{ij} \mid j \in \mathcal{I}_{l}\}$ yields an unbiased estimator of $\{A_{ij} \mid j \in \mathcal{I}_{l}\}$ and can be described by $|\mathcal{I}_{l}| = b^{*}$ bits.

**Privatization**   Client $i$ then perturbs the $b^{*}$-bit message $\{Q_{ij} \mid j \in \mathcal{I}_{l}\}$ into $\left\{\hat{Q}_{ij} \mid j \in \mathcal{I}_{l}\right\}$ via $2^{b^{*}}$-RR, as described in (8). Similarly,

$$\left\{\left(\frac{e^{\varepsilon} + 2^{b^{*}} - 1}{e^{\varepsilon} - 1}\right)\hat{Q}_{ij} \mid j \in \mathcal{I}_{l}\right\}$$

yields an unbiased estimator on $\{A_{ij} \mid j \in \mathcal{I}_{l}\}$.

**Estimation and the $\ell_{2}$ error**   For all $j \in \mathcal{I}_{l}$, $\hat{A}_{ij} \triangleq \left(\frac{e^{\varepsilon}+2^{b^{*}}-1}{e^{\varepsilon}-1}\right)\hat{Q}_{ij}$ yields an unbiased estimator on $\mathbb{E}_{Q}[A_{ij}]$, and note that $\hat{Q}_{ij} \in \left[-\frac{K}{\sqrt{d}}, \frac{K}{\sqrt{d}}\right]$, so the variance of $\hat{A}_{ij}$ is controlled by

$$\mathbb{E}_{Q}\left[\left(\hat{A}_{ij} - \theta\left(Q\right)\left(j\right)\right)\right] \leq \left(\frac{e^{\varepsilon} + 2^{b^{*}} - 1}{e^{\varepsilon} - 1}\right)^{2}\left(\frac{2K}{\sqrt{d}}\right)^{2} = O\left(\frac{1}{d\min\left(1, \varepsilon^{2}\right)}\right).$$

Since for each coordinate $j \in \mathcal{I}_{l}$, there are $|\mathcal{G}_{l}|$ clients (samples) that output independent and unbiased estimators $\hat{A}_{ij}$, the estimator

$$\hat{A}_{j} \triangleq \frac{1}{|\mathcal{G}_{l}|}\sum_{i \in \mathcal{G}_{l}}\hat{A}_{ij}$$

has variance

$$O\left(\frac{1}{d\,|\mathcal{G}_{l}|}\right) = O\left(\frac{1}{n\min\left(b^{*}, \varepsilon^{2}\right)}\right).$$

Therefore, we arrive at

$$\mathbb{E}\left[\sum_{j=1}^{N}\left(\hat{A}_{j} - \mathbb{E}_{Q}[A_{j}]\right)^{2}\right] = O\left(\frac{d}{n\min\left(b^{*}, \varepsilon^{2}\right)}\right).$$

Write $\hat{\theta} = \sum_{j=1}^{N}\hat{A}_{j}u_{j}$ and note that $\theta\left(P\right) = \sum_{j=1}^{N}\mathbb{E}_{Q}\left[\hat{A}_{j}\right]u_{j}$, so by (4) we conclude that

$$\mathbb{E}_{P}\left[\|\hat{\theta} - \theta(P)\|_{2}^{2}\right] = O\left(\frac{d}{n\min\left(b^{*}, \varepsilon^{2}\right)}\right) = O\left(\frac{d}{n\min\left(\varepsilon, \varepsilon^{2}, b\right)}\right).$$

∎

# E   Proof of Theorem 3.1

## E.1   Achieving optimal $\ell_{1}$ and $\ell_{2}$ error (part (i) of Theorem 3.1)

In this section, we show that Recursive Hadamard Response (RHR) achieves optimal $\ell_{1}$ and $\ell_{2}$ estimation error.

**Decomposition of Hadamard matrix** Let us set $B = d/2^{k-1}$. Since $H_d = H_{2^{k-1}} \otimes H_B$, for any $j \in [B]$ and $m \in [2^{k-1}]$, if $j' = (m-1)B + j$ (and thus $j \equiv j' \pmod{B}$), we must have $(H_d)_{j'} = (H_{2^{k-1}})_m \otimes (H_b)_j$, where $\otimes$ is the Kronecker product. This allows us to decompose the $j'$-th component of $H_d \cdot X_i$ into

$$(H_d)_{j'} \cdot X_i = ((H_{2^{k-1}})_m \otimes (H_B)_j) \cdot X_i = \sum_{l=1}^{2^{k-1}} (H_{2^{k-1}})_{m,l} (H_B)_j \cdot X_i^{(l)}, \qquad (13)$$

where $X_i^l$ is the $l$-th block of $X_i$, i.e. $X_i^{(l)} \triangleq X_i[(l-1)B + 1 : lB]$. Therefore, as long as we know $(H_B)_j \cdot X_i^{(l)}$ for $l = 1, ..., 2^{k-1}$, we can reconstruct $(H_d)_{j'} \cdot X_i$, for all $j' \equiv j \pmod{B}$.

**Encoding mechanism** Let $r_i \sim \text{Uniform}(B)$ be generated from the shared randomness, and consider the following quantizer

$$Q(X_i, r_i) = \left( (H_B)_{r_i} \cdot X_i^{(l)} \right)_{l=1,...,2^{k-1}} \in \{-1, 0, 1\}^{2^{k-1}}.$$

Since $X_i$ is one-hot encoded, there is exactly one non-zero $X_i^{(l)}$, so $Q(X_i, r_i)$ can be described by a $k$-bit string (with $k-1$ bits indicating the location of the non-zero entry and 1 bit indicating its sign).

Given $Q(X_i, r_i)$, by (13) we can recover $2^{k-1}$ coordinates of $Y_i = H_d \cdot X_i$:

$$Y_i(r') = (H_d)_{r'} \cdot X_i = \sum_{l=1}^{2^{k-1}} (H_{2^{k-1}})_{m,l} (H_B)_{r_i} \cdot X_i^{(l)} = (H_{2^{k-1}})_m \cdot Q(X_i, r_i), \qquad (14)$$

for any $r' = (m-1)B + r_i$. Therefore, if we define

$$\hat{Y}_i(Q(X_i, r_i), r_i) \triangleq \begin{cases} \frac{1}{2^{k-1}} Y_i(r'), & \text{if } r' \equiv r_i \\ 0, & \text{else,} \end{cases} \qquad (15)$$

then $\mathbb{E}\left[ \hat{Y}_i \right] = \frac{1}{d} H_d \cdot X_i$, where the expectation is taken with respect to $r_i$.

To protect privacy, client $i$ then perturbs $Q(X_i, r_i)$ via $2^k$-RR scheme, since $Q$ takes values on an alphabet of size $2^k$, denoted by $\mathcal{Q} = \{\pm e_1, \ldots, \pm e_{2^{k-1}}\}$,

$$\tilde{Q}_i = \begin{cases} Q(X_i, r_i), & \text{w.p. } \frac{e^\varepsilon}{e^\varepsilon + 2^k - 1} \\ Q' \in \mathcal{Q} \setminus \{Q(X_i, r_i)\}, & \text{w.p. } \frac{1}{e^\varepsilon + 2^k - 1}, \end{cases}$$

where $e_l$ denotes the $l$-th coordinate vector in $\mathbb{R}^{2^{k-1}}$.

Client $i$ then sends the $k$-bit report $\tilde{Q}_i$ to the server, and with $\tilde{Q}_i$, the server can compute an estimate of $Q_i$ since $\mathbb{E}\left[ \tilde{Q}_i \middle| Q(X_i, r_i) \right] = \frac{e^\varepsilon - 1}{e^\varepsilon + 2^k - 1} Q(X_i, r_i)$.

**Constructing estimator for $\hat{D}$** For a given $\tilde{Q}_i$, we estimate $Y_i$ by $\hat{Y}_i \left( \frac{e^\varepsilon + 2^k - 1}{e^\varepsilon - 1} \tilde{Q}_i, r_i \right)$, where $\hat{Y}_i$ is given by (14) and (15), with $Q(X_i, r_i)$ in (14) replaced by $\tilde{Q}_i$.

**Claim E.1** $\hat{Y}_i$ is an unbiased estimator of $Y_i$.

The final estimator of $D_{X^n} = \frac{1}{n} \sum X_i$ is given by

$$\hat{D}\left( \left( \tilde{Q}_i, r_i \right)_{i=1,...,n} \right) \triangleq \frac{1}{n} \sum_{i=1}^{n} H_d \cdot \hat{Y}_i \left( \frac{e^\varepsilon + 2^k - 1}{e^\varepsilon - 1} \tilde{Q}_i, r_i \right). \qquad (16)$$

Note that by Claim E.1, $\hat{D}$ is an unbiased estimator for $D_{X^n}$. Finally picking $k = \min\left( b, \lceil \varepsilon \log_2 e \rceil, \lfloor \log d \rfloor \right)$ yields the following bounds.

**Claim E.2** *The estimator $\hat{D}$ in (16) achieves the optimal $\ell_1$ and $\ell_2$ errors:*

$$\mathbb{E}\left[\left\|\hat{D} - D_{X^n}\right\|_2^2\right] \preceq \frac{d}{n\left(\min\left\{e^\varepsilon, (e^\varepsilon - 1)^2, 2^b, d\right\}\right)} \quad and$$

$$\mathbb{E}\left[\left\|\hat{D} - D_{X^n}\right\|_1\right] \preceq \frac{d}{\sqrt{n\left(\min\left\{e^\varepsilon, (e^\varepsilon - 1)^2, 2^b, d\right\}\right)}}.$$

This establishes the achievability part of Theorem 3.1. $\qquad\qquad\qquad\qquad\qquad\square$

## E.2 Algorithms

We summarize our proposed scheme RHR scheme below:

---
**Algorithm 1:** Encoding mechanism $\tilde{Q}_i$ (at each client)
---
**Input:** client index $i$, observation $X_i$, privacy level $\varepsilon$, alphabet size $d$
**Result:** Encoded message $\left(\tilde{\text{sign}}, \tilde{\text{loc}}\right)$
Set $D = 2^{\lceil \log d \rceil}$, $k = \min\left(b, \lceil \varepsilon \log_2 e \rceil\right)$, $B = D/2^{k-1}$;
Draw $r_i$ from uniform($B$) using public-coin ;
**begin**
    $\text{loc} \leftarrow \lceil \frac{X_i}{B} \rceil$;
    $\text{sign} \leftarrow (H_d)_{r_i, X_i}$;
    $\left(\tilde{\text{sign}}, \tilde{\text{loc}}\right) \leftarrow 2^k - \text{RR}_\varepsilon\left((\text{sign}, \text{loc})\right)$   /* (sign,loc) as a $k$-bit string */;
**end**

---

Notice that computing any entry of $H_d$ takes $O(\log d)$ Boolean operations, and uniformly sampling a $k$-bit string takes $O(k)$ time. Therefore the computation cost at each client is $O(\log d)$ time. Also note that the encoded message is a $k$-bit binary string, and therefore the communication cost at each client is $k = \min\left(b, \lceil \varepsilon \log_2(e) \rceil\right) \leq b$.

Once receiving the $k$-bit messages from all clients, the server does the following operation:

---
**Algorithm 2:** Estimator of $D_{X^n}$ (at the server)
---
**Input:** $(\tilde{\text{sign}}[1:n], \tilde{\text{loc}}[1:n])$, privacy level $\varepsilon$, alphabet size $d$
**Result:** $\hat{D}$
Set $D = 2^{\lceil \log d \rceil}$, $k = \min\left(b, \lceil \varepsilon \log_2 e \rceil\right)$, $B = D/2^{k-1}$;
Partition messages into groups $\mathcal{G}_1, ..., \mathcal{G}_B$, with message $i$ in $\mathcal{G}_{r_i}$;
**forall** $j = 1, ..., B$ **do**
    $\mathcal{G}_j^+ \leftarrow \left\{\tilde{\text{loc}}(i) \,|\, i \in \mathcal{G}_j, \tilde{\text{sign}}(i) = +1\right\}$;
    $\mathcal{G}_j^- \leftarrow \left\{\tilde{\text{loc}}(i) \,|\, i \in \mathcal{G}_j, \tilde{\text{sign}}(i) = -1\right\}$;
    $\text{Emp}_j \leftarrow \left(\text{empirical distribution}(\mathcal{G}_j^+) - \text{empirical distribution}(\mathcal{G}_j^-)\right) \cdot \frac{e^\varepsilon + 2^k - 1}{e^\varepsilon - 1}$;
    **forall** $l = 0, ..., 2^{k-1} - 1$ **do**
        $\hat{E}[l \cdot B + j] \leftarrow \text{FWHT}(\text{Emp}_j)[l]$    /* fast Walsh-Hadamard transform */
    **end**
**end**
$\hat{D} \leftarrow \frac{1}{d} \cdot \text{FWHT}\left(\hat{E}\right)$;

---

The encoding mechanism above involves two operations: 1) sampling a random index $r_i$ from $[B]$ at each client with the help of a public coin, and 2) computing $(H_d)_{r_i} \cdot X_i$. Since $X_i$ is one-hot, the encoding complexity is $O(\log d)$. On the other hand, in order to efficiently decode, the server first computes the joint histogram of client $i$'s report and $r_i$ in $O(n)$ time, which in turn allows us to calculate $\frac{1}{n}\sum_i \hat{Y}_i$, and then apply the Fast Walsh-Hadamard transform (FWHT) to obtain the estimator of empirical frequency in $O(d \log d)$ time. Hence the overall decoding complexity is $O(n + d\log d)$.

### E.3 Lower Bound on $\ell_1$ and $\ell_2$ errors in Theorem 3.1

We can bound the error by considering the worst case Bayesian setting, i.e. by imposing a prior distribution $\boldsymbol{p}$ on $X_1, ..., X_n$ and applying the converse part of Theorem 3.2 in Section 3.2.

Let $X_1, ..., X_n \overset{\text{i.i.d.}}{\sim} \boldsymbol{p}$. Then for any $\hat{D}(X^n)$, we must have

$$
\begin{aligned}
\max_{X^n \sim \boldsymbol{p}} \mathbb{E}\left[\left\|\hat{D} - D_{X^n}\right\|_2^2\right] &\overset{(a)}{\geq} \max_{\boldsymbol{p}} \mathbb{E}\left[\left(\left\|\hat{D} - \boldsymbol{p}\right\|_2 - \|D_{X^n} - \boldsymbol{p}\|_2\right)^2\right] \\
&\geq \max_{\boldsymbol{p}} \left(\mathbb{E}\left[\left\|\hat{D} - \boldsymbol{p}\right\|_2^2\right] - 2\mathbb{E}\left[\left\|\hat{D} - \boldsymbol{p}\right\|_2 \|D_{X^n} - \boldsymbol{p}\|_2\right]\right) \\
&\overset{(b)}{\geq} \max_{\boldsymbol{p}} \left(\mathbb{E}\left[\left\|\hat{D} - \boldsymbol{p}\right\|_2^2\right] - 2\sqrt{\mathbb{E}\left[\left\|\hat{D} - \boldsymbol{p}\right\|_2^2\right]\mathbb{E}\left[\|D_{X^n} - \boldsymbol{p}\|_2^2\right]}\right)
\end{aligned}
\tag{17}
$$

where (a) and (b) follow from the triangular inequality and the Cauchy-Schwarz inequality respectively. By Theorem 3.2, there exists a worst case $\boldsymbol{p}^*$ such that

$$
c\frac{d}{n}\left(\frac{1}{\min\left\{e^\varepsilon, (e^\varepsilon - 1)^2, 2^b\right\}}\right) \leq \mathbb{E}\left[\left\|\hat{D} - \boldsymbol{p}^*\right\|_2^2\right] \leq C\frac{d}{n}\left(\frac{1}{\min\left\{e^\varepsilon, (e^\varepsilon - 1)^2, 2^b\right\}}\right), \tag{18}
$$

for some constants $c$ and $C$. On the other hand, the $\ell_2$ convergence of $D(X^n)$ to $\boldsymbol{p}$ is $O(1/n)$ for any $\boldsymbol{p}$, which gives us

$$
\mathbb{E}\left[\|D_{X^n} - \boldsymbol{p}^*\|_2^2\right] \leq c'\frac{1}{n}. \tag{19}
$$

Plugging (18) and (19) back into (17) yields

$$
\begin{aligned}
&\max_{X^n \sim \boldsymbol{p}} \mathbb{E}\left[\left\|\hat{D} - D_{X^n}\right\|_2^2\right] \\
&\geq C_1\frac{d}{n}\left(\frac{1}{\min\left\{e^\varepsilon, (e^\varepsilon - 1)^2, 2^b\right\}}\right) - C_2\frac{1}{n}\sqrt{\frac{d}{\min\left\{e^\varepsilon, (e^\varepsilon - 1)^2, 2^b\right\}}}.
\end{aligned}
$$

Thus as long as $\min\left(e^\varepsilon, (e^\varepsilon - 1)^2, 2^b\right) = o(d)$, the first term dominates and the desired $\ell_2$ lower bound follows.

For the case of $\ell_1$, we similarly have

$$
\max_{X^n \sim \boldsymbol{p}} \mathbb{E}\left[\left\|\hat{D} - D_{X^n}\right\|_1\right] \geq \max_{\boldsymbol{p}} \left(\mathbb{E}\left[\left\|\hat{D} - \boldsymbol{p}\right\|_1\right] - \mathbb{E}\left[\|D_{X^n} - \boldsymbol{p}\|_1\right]\right) \tag{20}
$$

It is well-known that $\mathbb{E}\left[\|D(X^n) - \boldsymbol{p}\|_1\right] \leq \sqrt{d/n}$ (for instance, see [26]), and by the converse part of Theorem 3.2

$$
\max_{\boldsymbol{p}} \mathbb{E}\left[\left\|\hat{D} - \boldsymbol{p}\right\|_1\right] \geq \sqrt{\frac{d^2}{n\min\left\{e^\varepsilon, (e^\varepsilon - 1)^2, 2^b\right\}}}.
$$

Plugging this into (20) yields the $\ell_1$ lower bound. $\qquad\square$

### E.4 Achieving optimal $\ell_\infty$ error (part (ii) of Theorem 3.1)

To obtain an upper bound on $\ell_\infty$ error, we extend the `TreeHist` protocol in [11], a 1-bit LDP heavy hitter estimation mechanism, to communicate $b$ bits and satisfy a desired privacy level $\varepsilon$. A simpler version of `TreeHist` protocol, which is not optimized for computational complexity, is as follows: we first perform Hadamard transform on $X_i$, and sample one random coordinate with public randomness $r_i$. The 1-bit message is then passed through a binary $\varepsilon$-LDP mechanism. We can show that from the perturbed outcomes, the server can construct an unbiased estimator of $X_i$ with bounded sub-Gaussian norm, and the $\ell_\infty$ error will be $O(\sqrt{\log d/n\varepsilon^2})$.

To extend this scheme to an arbitrary privacy regime and an arbitrary communication budget of $b$ bits, we independently and uniformly sample the Hadamard transform of $X_i$ for $k = \min(b, \lceil \varepsilon \rceil)$ times. Each 1-bit sample is then perturbed via a $\varepsilon'$-LDP mechanism with $\varepsilon' \triangleq \varepsilon/k$.

Note that under the distribution-free setting, the randomness comes only from the sampling and the privatization steps, so we could view each re-sampled and perturbed message as generated from a fresh new copy of $X_i$ since $X_i$ is not random. Equivalently, this boils down to a frequency estimation problem with $n' = nk$ clients and under $\varepsilon' = \varepsilon/k$ and gives us the $\ell_\infty$ error

$$O\left(\sqrt{\frac{\log d}{n'(\varepsilon')^2}}\right) = O\left(\sqrt{\frac{\log d}{n\min(\varepsilon^2, \varepsilon, b)}}\right).$$

Below we describe the details.

**Encoding mechanism**  Set $k = \min(b, \lceil \varepsilon \rceil)$. For each $X_i$, we randomly sample $(H_d)_{X_i}$ (i.e. the $X_i$-th column of $H_d$) $k$ times, identically and independently by using the shared randomness. Let $r_i^{(1)}, ..., r_i^{(k)}$ be the sampled coordinates, which are known to both the server and node $i$, and $(H_d)_{X_i, r_i^{(\ell)}}$ be the sampling outcomes. Then due to the orthogonality of $H_d$, for all $j \in [d], \ell \in [k]$,

$$\mathbb{E}\left[(H_d)_{j, r_i^{(\ell)}} \cdot (H_d)_{X_i, r_i^{(\ell)}}\right] = \begin{cases} 1, & \text{if } j = X_i \\ 0, & \text{if } j \neq X_i, \end{cases} \tag{21}$$

where the expectation is taken over $r_i^{(\ell)}$.

We then pass $\left\{(H_d)_{X_i, r_i^{(\ell)}} \middle| \ell = 1, ..., k\right\}$ through $k$ binary $\varepsilon'$-LDP channels sequentially, with $\varepsilon' \triangleq \varepsilon/k$. By the composition theorem of differential privacy, the privatized outcomes, denoted as $\left\{(\tilde{H}_d)_{X_i, r_i^{(\ell)}}\right\}$, satisfy $\varepsilon$-LDP.

**Estimation of $D_{X^n}$**  Observe that

$$\mathbb{E}\left[\left(\frac{e^{\varepsilon'}+1}{e^{\varepsilon'}-1}\right)(\tilde{H}_d)_{X_i, r_i^{(\ell)}} \middle| (H_d)_{X_i, r_i^{(\ell)}}\right] = (H_d)_{X_i, r_i^{(\ell)}},$$

where the expectation is with respect to the privatization. Therefore

$$\hat{X}_i^{(\ell)}(j) \triangleq \left(\frac{e^{\varepsilon'}+1}{e^{\varepsilon'}-1}\right)(H_d)_{j, X_i}(\tilde{H}_d)_{X_i, r_i^{(\ell)}}$$

defines an unbiased estimator of $X_i(j)$. Moreover,

$$\left|\hat{X}_i^{(\ell)}(j) - X_i(j)\right| \leq \left(\frac{e^{\varepsilon'}+1}{e^{\varepsilon'}-1}+1\right) \text{ a.s.},$$

so $\hat{X}_i^{(\ell)}(j)$ has sub-Gaussian norm bounded by

$$\sigma \leq 2\frac{e^{\varepsilon'}+1}{e^{\varepsilon'}-1}. \tag{22}$$

Finally, we estimate $D_{X^n}(j)$ by

$$\hat{D}(j) = \frac{1}{nk}\sum_{i=1}^{n}\sum_{\ell=1}^{k}\hat{X}_i^{(\ell)}(j).$$

Observe that

$$\hat{D}(j) - D_{X^n}(j) = \frac{1}{nk}\sum_{i=1}^{n}\sum_{\ell=1}^{k}\left(\hat{X}_i^{(\ell)}(j) - X_i(j)\right) \tag{23}$$

has sub-Gaussian norm bounded by $\sigma/\sqrt{nk}$, where $\sigma$ is given by (22).

To bound the $\ell_\infty$ norm, we apply the maximum bound (see, for instance, [43, Chapter 2]) for sub-Gaussian random variables (note that for $j, j'$, $\hat{D}(j)$ and $\hat{D}(j')$ are not independent):

$$\mathbb{E}\left[\max_{j \in [d]} \left|\hat{D}(j) - D_{X^n}(j)\right|\right] \leq 2\sqrt{\sigma^2 \log d} = 4\sqrt{\left(\frac{e^{\varepsilon'} + 1}{e^{\varepsilon'} - 1}\right)^2 \frac{\log d}{nk}} \overset{(a)}{\asymp} \sqrt{\frac{\log d}{n \min\left(\varepsilon, \varepsilon^2, k\right)}}, \quad (24)$$

where (a) holds since if $\varepsilon = o(1)$, then $k = 1$ and hence

$$\left(\frac{e^{\varepsilon'} + 1}{e^{\varepsilon'} - 1}\right)^2 \asymp \frac{1}{\varepsilon^2};$$

otherwise $\varepsilon = \Omega(1)$ and $\varepsilon' = \Omega(1)$, so

$$\left(\frac{e^{\varepsilon'} + 1}{e^{\varepsilon'} - 1}\right)^2 \asymp 1.$$

Both cases are upper bounded by (24), so the result follows. $\qquad\square$

**Remark E.1** *Notice that in the high privacy regime $\varepsilon = o(1)$, the upper bound matches the lower bound in [12]. For general privacy regimes with limited communication, however, we do not know whether the upper bound is tight or not. This remains as an open question.*

## F  Proof of Theorem 3.2

The construction of the distribution estimation scheme mainly follows Section E.1, except we replace the random sampling step by a deterministic grouping idea. We will use the same notation as in Section E.1.

**Encoding mechanism**  We group $n$ samples into $B$ equal-sized groups, each with $n' = n/B$ samples. For sample $X_i \in \mathcal{G}_j$, we quantize it to a $2^{k-1}$-dimensional $\{1, 0, -1\}$ vector:

$$Q_j(X_i) = \begin{bmatrix} (H_B)_j \cdot X_i^{(1)} \\ (H_B)_j \cdot X_i^{(2)} \\ \vdots \\ (H_B)_j \cdot X_i^{(2^{k-1})} \end{bmatrix} \in \{-1, 0, 1\}^{2^{k-1}}.$$

Since $X_i$ is one-hot encoded, there is only one $l \in \{1, ..., 2^{k-1}\}$ such that $(H_B)_j \cdot X_i^{(l)} \neq 0$, so $Q_j(X_i)$ can be described by $k$ bits (1 bit for the sign and $(k-1)$ bits for the location of the non-zero element). Also notice that

$$\mathbb{E}\left[Q_j(X_i)\right] = \begin{bmatrix} (H_B)_j \cdot \boldsymbol{p}^{(1)} \\ (H_B)_j \cdot \boldsymbol{p}^{(2)} \\ \vdots \\ (H_B)_j \cdot \boldsymbol{p}^{(2^{k-1})} \end{bmatrix},$$

where $\boldsymbol{p}^{(l)} \triangleq \boldsymbol{p}[(l-1)B + 1 : lB]$. By (13), the estimator $\hat{q}_{j'} = \langle (H_{2^{k-1}})_m, Q_j(X_i) \rangle$ is unbiased for $q_{j'}$ (where $j' = (m-1)B + j$).

We further perturb $Q_j$ via $2^k$-RR scheme, since $Q$ takes values on an alphabet of size $2^k$, denoted by $\mathcal{Q} = \{\pm e_1, \ldots, \pm e_{2^{k-1}}\}$,

$$\tilde{Q}_j = \begin{cases} Q_j, \text{ w.p. } \frac{e^\varepsilon}{e^\varepsilon + 2^k - 1} \\ Q' \in \mathcal{Q} \setminus \{Q_j\}, \text{ w.p. } \frac{1}{e^\varepsilon + 2^k - 1}, \end{cases}$$

where $e_l$ denotes the $l$-th coordinate vector in $\mathbb{R}^{2^{k-1}}$. This gives us

$$\mathbb{E}\left[\tilde{Q}_j\right] = \frac{e^\varepsilon - 1}{e^\varepsilon + 2^k - 1} \mathbb{E}\left[Q_j\right].$$

Therefore $\frac{e^\varepsilon + 2^k - 1}{e^\varepsilon - 1} \tilde{Q}_j$ yields an unbiased estimator of

$$
\begin{bmatrix}
(H_B)_j \cdot \boldsymbol{p}^{(1)} \\
(H_B)_j \cdot \boldsymbol{p}^{(2)} \\
\vdots \\
(H_B)_j \cdot \boldsymbol{p}^{(2^{k-1})}
\end{bmatrix}.
$$

**Constructing the estimator for $p$**    For each $j' \equiv j \pmod{B}$, we estimate $(H_{2^{k-1}})_m \cdot Q_j(X_i), i \in \mathcal{G}_j$ (recall that $j' = j + (m-1)B$). Define the estimator

$$
\hat{q}_{j'} \left( \{X_i, i \in \mathcal{G}_j\} \right) = \frac{1}{|\mathcal{G}_j|} \sum_{i \in \mathcal{G}_j} (H_{2^{k-1}})_m \cdot \left( \frac{e^\varepsilon + 2^k - 1}{e^\varepsilon - 1} \right) \tilde{Q}_j(X_i)
$$

$$
= \frac{B}{n} \left( \frac{e^\varepsilon + 2^k - 1}{e^\varepsilon - 1} \right) \sum_{i \in \mathcal{G}_j} (H_{2^{k-1}})_m \tilde{Q}_j(X_i).
$$

The MSE of $\hat{q}_{i'}$ can be obtained by

$$
\mathbb{E} \left[ (\hat{q}_{j'} - q_{j'})^2 \right] \overset{(a)}{=} \mathsf{Var}\,(\hat{q}_{i'})
$$

$$
\overset{(b)}{=} \frac{d}{n2^{k-1}} \left( \frac{e^\varepsilon + 2^k - 1}{e^\varepsilon - 1} \right)^2 \mathsf{Var}\left( (H_{2^{k-1}})_m \cdot \tilde{Q}_j(X_i) \right)
$$

$$
\overset{(c)}{\leq} \frac{d}{n2^{k-1}} \left( \frac{e^\varepsilon + 2^k - 1}{e^\varepsilon - 1} \right)^2, \tag{25}
$$

where (a) is due to the unbiasedness of $\hat{q}_{j'}$, (b) is due to the independence across $X_i$, and (c) is because $\langle (H_{2^{k-1}})_m, \tilde{Q}_j \rangle$ only takes value in $\{-1, 1\}$.

Finally, let $\hat{\boldsymbol{p}}$ be the inverse Hadamard transform of $\hat{\boldsymbol{q}}$, the MSE is

$$
\mathbb{E} \|\hat{\boldsymbol{p}} - \boldsymbol{p}\|_2^2 = \mathbb{E} \left[ \langle \hat{\boldsymbol{p}} - \boldsymbol{p}, \hat{\boldsymbol{p}} - \boldsymbol{p} \rangle \right]
$$

$$
= \mathbb{E} \left[ (\hat{\boldsymbol{q}} - \boldsymbol{q})^\intercal \left( H_d^{-1} \right)^\intercal H_d^{-1} (\hat{\boldsymbol{q}} - \boldsymbol{q}) \right]
$$

$$
= \frac{1}{d} \mathbb{E} \|\hat{\boldsymbol{q}} - \boldsymbol{q}\|_2^2
$$

$$
\leq \frac{d}{n2^k} \left( \frac{e^\varepsilon + 2^k - 1}{e^\varepsilon - 1} \right)^2
$$

$$
= O \left( \frac{d}{n2^k} \left( \frac{e^\varepsilon + 2^k}{e^\varepsilon - 1} \right)^2 \right),
$$

where the last inequality holds due to (25).

Picking $k = \min \left( b, \lceil \varepsilon \log_2 e \rceil, \lfloor \log d \rfloor \right)$ yields

$$
\mathbb{E} \|\hat{\boldsymbol{p}} - \boldsymbol{p}\|_2^2 = O \left( \frac{d}{n \min (2^b, e^\varepsilon, d)} \left( \frac{e^\varepsilon}{e^\varepsilon - 1} \right)^2 \right).
$$

Observe that if $e^\varepsilon = O(2^b)$, then $e^\varepsilon \preceq 2^b$, so $\mathbb{E} \|\hat{\boldsymbol{p}} - \boldsymbol{p}\|_2^2 = O \left( \frac{de^\varepsilon}{n(e^\varepsilon - 1)^2} \right)$. On the other hand, if $e^\varepsilon = \Omega(2^b)$, then $\frac{e^\varepsilon}{e^\varepsilon - 1} = \theta(1)$, and $\mathbb{E} \|\hat{\boldsymbol{p}} - \boldsymbol{p}\|_2^2 = O \left( \frac{d}{n \min(2^b, d)} \right)$.

Therefore we conclude that

$$
\mathbb{E} \|\hat{\boldsymbol{p}} - \boldsymbol{p}\|_2^2 \preceq \max \left( \frac{d}{n \min (2^b, d)}, \frac{de^\varepsilon}{n(e^\varepsilon - 1)^2} \right) \asymp \frac{d}{n} \left( \frac{1}{\min \left\{ e^\varepsilon, (e^\varepsilon - 1)^2, 2^b, d \right\}} \right).
$$

Finally, by Jensen's inequality and Cauchy-Schwarz inequality, we also have

$$\mathbb{E}\left[\|\hat{\boldsymbol{p}} - \boldsymbol{p}\|_1\right] \le \left(\mathbb{E}\left[\|\hat{\boldsymbol{p}} - \boldsymbol{p}\|_1^2\right]\right)^{\frac{1}{2}} \le \left(d \cdot \mathbb{E}\|\hat{\boldsymbol{p}} - \boldsymbol{p}\|_2^2\right)^{\frac{1}{2}} \preceq \frac{d}{\sqrt{n\left(\min\left\{e^\varepsilon, (e^\varepsilon - 1)^2, 2^b, d\right\}\right)}},$$

establishing the achievability part of Theorem 3.2. $\qquad\square$

### F.1 Algorithms and analysis

Each client runs the following algorithm:

---
**Algorithm 3:** Encoding mechanism (at each client)
---
**Input:** client index $i$, observation $X_i$, privacy level $\varepsilon$, alphabet size $d$
**Result:** Encoded message $(\tilde{\mathtt{sign}}, \tilde{\mathtt{loc}})$
Set $D = 2^{\lceil \log d \rceil}$. Set $k = \min(b, \lceil \varepsilon \log_2 e \rceil)$, $B = D/2^{k-1}$;
**begin**
    $j \leftarrow i \mod B$          /* assign user $i$ to group $j$ */;
    $\mathtt{loc} \leftarrow \lceil \frac{X_i}{B} \rceil$;
    $\mathtt{sign} \leftarrow (H_d)_{j, X_i}$;
    $(\tilde{\mathtt{sign}}, \tilde{\mathtt{loc}}) \leftarrow k\text{RR}_\varepsilon((\mathtt{sign}, \mathtt{loc}))$;
**end**

---

As in Algorithm 1, the computation cost at each client is $O(\log d)$. Also note that the encoded message is a $k$-bit binary string, and therefore the communication cost at each client is $k = \min(b, \varepsilon \log_2(e)) \le b$.

Upon receiving the privatized $k$-bit messages from the clients, the server runs the following algorithm:

---
**Algorithm 4:** Estimation of $\boldsymbol{p}$ (at the server)
---
**Input:** $(\tilde{\mathtt{sign}}[1:n], \tilde{\mathtt{loc}}[1:n])$, privacy level $\varepsilon$, alphabet size $d$
**Result:** $\hat{\boldsymbol{p}}$
Set $D = 2^{\lceil \log d \rceil}$, $k = \min(b, \lceil \varepsilon \log_2 e \rceil)$, $B = D/2^{k-1}$;
Partition messages into groups $\mathcal{G}_1, ..., \mathcal{G}_B$, with message $i$ in $\mathcal{G}_j$ if $i \equiv j \pmod{B}$;
**forall** $j = 1, ..., B$ **do**
    $\mathcal{G}_j^+ \leftarrow \{\tilde{\mathtt{loc}}(i) \mid i \in \mathcal{G}_j, \tilde{\mathtt{sign}}(i) = +1\}$;
    $\mathcal{G}_j^- \leftarrow \{\tilde{\mathtt{loc}}(i) \mid i \in \mathcal{G}_j, \tilde{\mathtt{sign}}(i) = -1\}$;
    $D_j \leftarrow (\text{empirical distribution}(\mathcal{G}_j^+) - \text{empirical distribution}(\mathcal{G}_j^-)) \cdot \frac{e^\varepsilon + 2^k - 1}{e^\varepsilon - 1}$;
    **forall** $l = 0, ..., 2^{k-1} - 1$ **do**
        $\hat{\boldsymbol{q}}[l \cdot B + j] \leftarrow \text{FWHT}(D_j)[l]$;
    **end**
**end**
$\hat{\boldsymbol{p}} \leftarrow \frac{1}{d} \cdot \text{FWHT}(\hat{\boldsymbol{q}})$;

---

Partitioning $n$ samples into $B$ groups and computing the empirical distribution of each group takes $O(n)$ time, and the fast Walsh-Hadamard transform can be performed in $O(d \log d)$ time. Hence the decoding complexity is $O(n + d \log d)$.

# G  Proofs for Section B

We start with proving Lemma B.1. Without access to the public randomness, [3] shows that at least $\Theta(d)$ bits of communication is required for heavy hitter estimation in order to obtain a consistent estimator[5]. We state their result here:

**Lemma G.1 ( [3] Theorem 4)** *Let $b \leq \log d - 2$. For all private-coin schemes $\left(Q^n, \hat{D}\right)$ with only private randomness and $b$ bits communication budgets, there exists a data sets $X_1, ..., X_n$ with $n > 12(2^b + 1)^2$, such that*

$$\mathbb{E}\left[\left\|\hat{D}(Q^n) - D_{X^n}\right\|_\infty\right] \geq \frac{1}{2^{b+2} + 4}.$$

Based on this, we claim that without public coin, each client needs to transmit at least $\Theta(\log d)$ bits in order to construct consistent schemes for frequency estimation or mean estimation.

## G.1  Proof of Lemma B.1

**Frequency estimation**    We lower bound $\ell_1$ and $\ell_2$ error by $\ell_\infty$ and apply Lemma G.1.

$$\mathbb{E}\left[\left\|\hat{D}(Q^n) - D_{X^n}\right\|_1\right] \geq \mathbb{E}\left[\left\|\hat{D}(Q^n) - D_{X^n}\right\|_\infty\right] \geq \frac{1}{2^{b+2} + 4},$$

and

$$\begin{aligned}
\mathbb{E}\left[\left\|\hat{D}(Q^n) - D_{X^n}\right\|_2^2\right] &\geq \mathbb{E}\left[\left\|\hat{D}(Q^n) - D_{X^n}\right\|_\infty^2\right] \\
&\geq \left(\mathbb{E}\left[\left\|\hat{D}(Q^n) - D_{X^n}\right\|_\infty\right]\right)^2 \\
&\geq \left(\frac{1}{2^{b+2} + 4}\right)^2.
\end{aligned} \tag{26}$$

This implies that it is impossible to construct consistent schemes with less than $\log d - 2$ bits per client in the absence of a public randomness. On the other hand, given $\log d$ bits, one can readily achieve the optimal estimation accuracy without any public randomness, for instance, by using Hadamard response [4] (see also the discussion in [3]). Therefore, the problem of frequency estimation is somewhat trivialized in the absence of public randomness.

**Mean estimation**    Let $X_i \in [d]$ be one-hot encoded, so $X_i \in \mathcal{B}_d(\mathbf{0}, 1)$. Then (26) implies the $\ell_2$ error of mean estimation is at least $1/\left(2^{b+2} + 4\right)^2$. Thus with less than $\log d - 2$ bits of communication budget, it is also impossible to construct a consistent scheme for mean estimation. $\square$

## G.2  Proof of Corollary B.1 and Corollary B.1

Notice that since one can always "simulate" the public coin by uplink communication (i.e. each client generates its private random bits and send them to the server), any $b$ bits public-coin scheme can be cast into a private coin scheme with additional $b$ bits communication. This implies the above impossibility results (Lemma B.1) also serves a valid lower bound for the amount of public randomness: for any public-coin scheme with $b < \log d - 2$ bits communication budgets, we need at least $\log d - b - 2$ bits of shared randomness in order to obtain a consistent estimate of the empirical mean or empirical frequency. $\square$

# H  Proof of Claims

## H.1  Proof of Claim D.1

**Proof.** According to (4), it suffices to control $\mathsf{Var}\left(\hat{a}_j\right)$. To bound the variance, consider

$$
\begin{aligned}
\mathsf{Var}\left(\hat{a}_j\right) &= \frac{N^2}{k^2} \cdot \left(\frac{e^\varepsilon + 2^k - 1}{e^\varepsilon - 1}\right)^2 \mathsf{Var}\left(\sum_{m=1}^{k} \tilde{q}_m \cdot \mathbb{1}_{\{j=s_m\}}\right) \\
&\leq \frac{N^2}{k^2} \cdot \left(\frac{e^\varepsilon + 2^k - 1}{e^\varepsilon - 1}\right)^2 \mathbb{E}\left[\left(\sum_{m=1}^{k} \tilde{q}_m \cdot \mathbb{1}_{\{j=s_m\}}\right)^2\right] \\
&\overset{(a)}{\leq} \frac{N^2}{k^2} \cdot \left(\frac{e^\varepsilon + 2^k - 1}{e^\varepsilon - 1}\right)^2 \left(\frac{c}{\sqrt{d}}\right)^2 \mathbb{E}\left[\left(\sum_{m=1}^{k} \mathbb{1}_{\{j=s_m\}}\right)^2\right] \\
&\overset{(b)}{\leq} C\frac{N}{k^2} \cdot \left(\frac{e^\varepsilon + 2^k - 1}{e^\varepsilon - 1}\right)^2 \left(\frac{k^2}{N^2} + \frac{k}{N}\right) \\
&= C\left(\frac{e^\varepsilon + 2^k - 1}{e^\varepsilon - 1}\right)^2 \left(\frac{1}{N} + \frac{1}{k}\right),
\end{aligned}
$$

where (a) is due to $|\tilde{q}_m| = \frac{c}{\sqrt{d}}$, and (b) is due to the second moment bound on $\mathsf{Binomial}(k, 1/N)$ and the fact $N = \Theta(d)$. Therefore by (4),

$$
\mathbb{E}\left[\left\|\hat{X} - X\right\|_2^2\right] \leq C_0 \sum_{i=1}^{N} \mathsf{Var}\left(\hat{a}_i\right) \leq C_1 \left(\frac{e^\varepsilon + 2^k - 1}{e^\varepsilon - 1}\right)^2 \frac{d}{k},
$$

establishing the claim. ∎

## H.2  Proof of Claim E.1

**Proof.** $\hat{Y}_i$ yields an unbiased estimator since

$$
\begin{aligned}
\mathbb{E}\left[\hat{Y}_i\left(\frac{e^\varepsilon + 2^k - 1}{e^\varepsilon - 1}\tilde{Q}_i, r_i\right)\right] &= \mathbb{E}\left[\mathbb{E}\left[\hat{Y}_i\left(\frac{e^\varepsilon + 2^k - 1}{e^\varepsilon - 1}\tilde{Q}_i, r_i\right)\bigg| r_i\right]\right] \\
&\overset{(a)}{=} \mathbb{E}\left[\hat{Y}_i\left(\mathbb{E}\left[\frac{e^\varepsilon + 2^k - 1}{e^\varepsilon - 1}\tilde{Q}_i\bigg| r_i\right], r_i\right)\right] \\
&= \mathbb{E}\left[\hat{Y}_i\left(Q(X_i, r_i), r_i\right)\right] \\
&= \frac{1}{d}H_d X_i, \qquad\qquad\qquad (27)
\end{aligned}
$$

where (a) holds since conditioning on $r_i$, $\hat{Y}_i(Q, r_i)$ is a linear function of $Q$. ∎

## H.3  Proof of Claim E.2

**Proof.** The $\ell_2$ error is

$$
\begin{aligned}
\mathbb{E}\left[\left\|\hat{D} - D_{X^n}\right\|_2^2\right] &= \frac{1}{n^2} \sum_{i=1}^{n} \mathbb{E}\left[\left\|H_d\hat{Y}_i - H_d\mathbb{E}\left[\hat{Y}_i\right]\right\|_2^2\right] \\
&= \frac{d}{n^2} \sum_{i=1}^{n} \mathbb{E}\left[\left\|\hat{Y}_i - \mathbb{E}\left[\hat{Y}_i\right]\right\|_2^2\right]. \qquad (28)
\end{aligned}
$$

It remains to bound $\mathbb{E}\left[\left\|\hat{Y}_i - \mathbb{E}\left[Y_i\right]\right\|_2^2\right]$. Observe that

$$
\left|\mathbb{E}[\hat{Y}_i]\right| = \left|\frac{H_d \cdot X_i}{d}\right| = [1/d, ..., 1/d]^\mathsf{T},
$$

and from expression (15), given $r_i$, there are only $2^{k-1}$ non-zero coordinates, each with value bounded by $\left(\frac{e^\varepsilon + 2^k - 1}{e^\varepsilon - 1}\right)/2^{k-1}$. Therefore we have

$$\mathbb{E}\left[\left\|\hat{Y}_i - \mathbb{E}\left[\hat{Y}_i\right]\right\|_2^2\right] = \mathbb{E}\left[\mathbb{E}\left[\left\|\hat{Y}_i - \mathbb{E}\left[\hat{Y}_i\right]\right\|_2^2 \Big| r_i\right]\right]$$
$$\leq 2\left(d\left(\frac{1}{d}\right)^2 + 2^{k-1}\left(\frac{e^\varepsilon + 2^k - 1}{2^{k-1}(e^\varepsilon - 1)}\right)^2\right).$$

Plugging this in to (28), we arrive at

$$\mathbb{E}\left[\left\|\hat{D} - D_{X^n}\right\|_2^2\right] \preceq \frac{d}{n2^{k-1}}\left(\frac{e^\varepsilon + 2^k - 1}{(e^\varepsilon - 1)}\right)^2.$$

Picking $k = \min\left(b, \lceil \varepsilon \log_2 e \rceil, \lfloor \log d \rfloor\right)$ yields

$$\mathbb{E}\left[\left\|\hat{D} - D_{X^n}\right\|_2^2\right] = O\left(\frac{d}{n\min\left(2^b, e^\varepsilon, d\right)}\left(\frac{e^\varepsilon}{e^\varepsilon - 1}\right)^2\right).$$

Observe that

(i) if $e^\varepsilon = O(2^b)$, then $e^\varepsilon \preceq 2^b$, so $\mathbb{E}\left[\left\|\hat{D} - D_{X^n}\right\|_2^2\right] = O\left(\frac{de^\varepsilon}{n(e^\varepsilon - 1)^2}\right)$.

(ii) If $e^\varepsilon = \Omega(2^b)$, then $\frac{e^\varepsilon}{e^\varepsilon - 1} = \theta(1)$, and $\mathbb{E}\left[\left\|\hat{D} - D_{X^n}\right\|_2^2\right] = O\left(\frac{d}{n\min(2^b, d)}\right)$.

Therefore we conclude that

$$\mathbb{E}\left[\left\|\hat{D} - D_{X^n}\right\|_2^2\right] \preceq \max\left(\frac{d}{n\min\left(2^b, d\right)}, \frac{de^\varepsilon}{n(e^\varepsilon - 1)^2}\right) \asymp \frac{d}{n}\left(\frac{1}{\min\left\{e^\varepsilon, (e^\varepsilon - 1)^2, 2^b, d\right\}}\right).$$

By Jensen's inequality and Cauchy-Schwarz inequality, we also have

$$\mathbb{E}\left[\left\|\hat{D} - D_{X^n}\right\|_1\right] \leq \left(\mathbb{E}\left[\left\|\hat{D} - D_{X^n}\right\|_1^2\right]\right)^{\frac{1}{2}} \leq \left(d \cdot \mathbb{E}\left\|\hat{D} - D_{X^n}\right\|_2^2\right)^{\frac{1}{2}}$$
$$\preceq \frac{d}{\sqrt{n\left(\min\left\{e^\varepsilon, (e^\varepsilon - 1)^2, 2^b, d\right\}\right)}}.$$

∎

## Footnotes

[1] A scheme is *consistent* if it has vanishing estimation error as $n \to \infty$.

[2]The definition of $\tilde{r}_{ME}(\cdot)$ is the same as that of $r_{ME}(\cdot)$ in (1), except that now the minimum is taken over all private-coin schemes.

[3]The code can be found in https://github.com/WeiNingChen/Kashin-mean-estimation (for the SQKR scheme) and https://github.com/WeiNingChen/RHR (for the RHR scheme).

[4]For HR, we use the codes from [4] (https://github.com/zitengsun/hadamard_response)

[5]Recall that an estimator is consistent if it has vanishing estimation error as $n$ tends to infinity.