[Reviews · NeurIPS 2020]

Review 1

Summary and Contributions: This paper studies fundamental problems of frequency estimation, distribution estimation, and mean estimation under local differential privacy. While these problems have been studied extensively, the prior work rarely ponders on the amount of communication (both from server in terms of common randomness and from the users). Bassily and Smith showed that using common randomness from the server, any LDP scheme can be converted into a one-bit communication scheme from each user. However, the necessity of common randomness was unclear, and so was the question of how much of it is needed? Acharya and Sun ([1] of the paper) showed that without common randomness for epsilon<1, one bit communication is enough to achieve optimal distribution estimation under L1 distance, and that it is impossible to do so for frequency estimation. However, for epsilon>1, the problem was not considered. This paper shows that as the value of epsilon changes, there are some interesting trade-offs that happen between sample complexity and the number of bits of communication per player. For example, for distribution estimation, they show that one bit protocols are not optimal for large epsilon, and one can actually use the additional communication in order to reduce the sample complexity of estimation. For the problem of frequency estimation, they show that as epsilon increases, the error can be reduced for larger epsilon. They also show a similar result for the problem of mean estimation, where the goal is to estimate the mean of vectors that are held at the users. This shows the wide range of the problems to which their arguments can generalize. In a nutshell, this paper shows that there is a trade-off between epsilon and communication for a given number of samples and error requirement. This result however is to be expected since with eps=log k communication is the only bottleneck that remains. The algorithms are simple, and are modifications to the previous approaches using Hadamard matrices. The approach has a potential of generalizing to other problems under LDP, a question that could be interesting to explore. The writing of the paper is clear in general, but perhaps the authors can provide some intuition as to why such a trade-off exists between epsilon and communication. ———————- Post rebuttal: I have read the author response and am updating my score accordingly.

Strengths: See above.

Weaknesses: See above.

Correctness: They look correct.

Clarity: Well written

Relation to Prior Work: Reasonably well discussed.

Reproducibility: Yes

Additional Feedback:


Review 2

Summary and Contributions: This paper studies several of the most basic problems in distributed statistical estimation: 1) frequency estimation, where the goal is to approximate the histogram of users' categorical data, 2) distribution estimation, which is similar to the above, but where users have i.i.d. samples from an unknown distribution and the goal is to approximate that distribution, and 3) mean estimation, where user data comes from a unit ball and the goal is to estimate the mean. These problems have been studied under various constraints on how data is communicated from individuals to an aggregator, including local privacy and limited communication, with tight upper and lower bounds known for each of those restrictions separately. This paper studies these problems under simultaneous privacy and communication constraints. That is, it aims to understand the optimal estimation error for communication-constrained locally private protocols. It gives asymptotically optimal protocols for each of these settings (under l_1, l_2, and l_infinity error for 1), l_2 and l_infinity error for 2), and l_2 error for 3)), and hold for the full range of privacy and communication parameters. Moreover, these protocols show that the less stringent of the privacy or communication requirement can be achieved for free, with no asymptotic increase in error. The new protocols for frequency and distribution estimation refine the "Hadamard response" algorithm from prior work. Hadamard response enables accurate frequency estimation for small epsilon with one bit of communication. The new protocol uses the recursive structure of Hadamard matrices to simultaneously increase communication and privacy loss, getting an optimal tradeoff for large epsilon. The new protocol for mean estimation uses a randomized response based on Kashin's encoding, which to my knowledge is novel in this line of work.

Strengths: This work considers some of the most basic problems in a very active area of research. It unifies a number of prior results and shows how to make them work for a full range of parameters, using clean new ideas. The results present a clear conceptual message, that at least in the settings considered, the less restrictive of privacy or low-communication can be had for free.

Weaknesses: The concrete new results concern the large epsilon (low privacy) regime where eps > 1. This is not such a desirable setting (which, in part, explains why it's been less explored in prior theoretical work); nevertheless, it is frequently used in applications.

Correctness: I did not review the proofs in detail, but the exposition of the techniques make the results appear to be correct.

Clarity: The paper is generally well-written. The tables do not do a great job of communicating which results are new and which ones follow from prior work, e.g., the Table 1 entry on Theorem 2.1. for heavy hitters. What do the blue and red backgrounds mean in Table 2?

Relation to Prior Work: There is an extensive literature on these problems from a number of different communities, and the discussion of related work seems to hit most of it. Something that is not clear is what results on mean estimations follow from (some version of) Duchi-Jordan-Wainwright. They consider mean estimation in the unit ball under minimax l_2 error, which should already have some implication for this problem.

Reproducibility: Yes

Additional Feedback: Is there an obstacle to getting a tight upper bound for distribution estimation under l_infinity error as well? EDIT: Thanks to the authors for their feedback. It confirms my thoughts about the paper and I'd be happy to see it accepted.


Review 3

Summary and Contributions: This paper studies the problems of frequency estimation and mean estimation under both privacy constraint and communication constraint, and proposes mechanisms that simultaneously achieve optimal privacy and communication efficiency using public randomness.

Strengths: For frequency estimation and mean estimation under LDP, the authors design optimal schemes using public randomness in terms of the trade-off between the communication cost and the accuracy. For distribution estimation under LDP, the authors show that private randomness suffices.

Weaknesses: I have doubts about the theoretical claims in this paper, especially on the lower bound, since [8] has shown that in the public coin model, every LDP protocol can be transformed so that each user sends only one bit to the central server with almost no loss in performance. Thus when using public randomness, the communication budget will not incur an accuracy loss for any LDP protocol by using the transformation in [8]. This is also discussed in [1], and therefore the authors of [1] study the converse question: Is public randomness necessary to reduce communication from users under LDP. ======== after reading the authors' response I upgrade the score to 6, as the result in [8] only applies to O(1) epsilon and this paper can complement it.

Correctness: See comments in the section of Weaknesses.

Clarity: This paper is well written and structurally clear.

Relation to Prior Work: This paper improves the error for the problem of mean estimation compared with the existing works under the same privacy budget and communication budget, and generalizes the achievability to arbitrary privacy budget and communication budget for the problem of frequency estimation.

Reproducibility: Yes

Additional Feedback:


Review 4

Summary and Contributions: This paper studies the limits on accuracy for a given privacy and communication budgets. The recursive Hadamard response (RHR) is presented for frequency estimation, where the estimation error decays while the privacy and communication budgets increase. In turn, from the RHR, a scheme for distribution estimation is proposed which requires less communication for the same accuracy and privacy levels in literature. Moreover, this paper also presents a scheme for mean estimation, which is based on the Kashin's sampling. ------------ Post rebuttal: I have read the author response, the authors have addressed my comments, and I stick to my score.

Strengths: The paper presents a novel idea which bridges the gap between the privacy constraints and the communication budget and achieves the same levels of privacy for lower communication cost. The given claims are supported and proven. The work is very significant to the NeurIPS audience.

Weaknesses: I think it would benefit the paper and the readers to discuss the shared randomness in more detail, since it is public. It would be good to discuss how and why this is added and how it affects privacy.

Correctness: The claims in this paper are correct. The paper satisfies its claims and gives sufficient proofs, explanations, and experimental results.

Clarity: The paper is in general well written, with some minor typos, like for example: - On page 2, section 1.1, "as follows" not "as follow" - On page 5, second paragraph, "optimality" not "optimility" It would be good to proofread the paper for similar typos. Moreover, the way tables 2 and 3 are places, it is hard to read the captions, maybe separate the two tables, or make them as subtables with more distance between them.

Relation to Prior Work: The relation of this work to the literature is discussed clearly and well.

Reproducibility: Yes

Additional Feedback:

[Author Response · NeurIPS 2020]

We thank the reviewers for their insightful comments. Below we respond to the main points raised by the reviewers.
The manuscript will be updated to reflect the reviewers' suggestions and our responses below.

**Reviewer 1:** Our results show that for any privacy budget $\varepsilon$, we only need $\lceil (\log_2 e) \cdot \varepsilon \rceil$ bits of communication budget
to achieve the order-optimal estimation error under $\varepsilon$-LDP. This result can be viewed in two different ways. First, as the
reviewer suggests, when we use larger $\varepsilon$, we need a larger communication budget to achieve the order-optimal error.
Second, as we emphasize in the paper, we cannot improve the error further by using a communication budget larger
than this characterized threshold, since the performance will be dominated by the more stringent constraint, i.e. privacy.

*Intuition behind our results.* The privacy level $\varepsilon$ dictates the "noise level" of the channel we induce from each client to
the server, and therefore its "capacity", the amount of information that can be supported by this channel. The larger $\varepsilon$,
the larger this capacity and therefore increasing the communication budget benefits the estimation task, but only up to
a certain threshold, i.e. the capacity of the $\varepsilon$-LDP channel. However, even with this intuition, we believe it is highly
surprising that the encoding over this channel can be done in such a way that it is simultaneously optimal from a both
privacy and compression perspective. While our algorithms include some elements from previous approaches, they
use these elements in novel ways. For example, as pointed out by the reviewer our scheme for frequency estimation
builds on Hadamard matrices, which also appears in [1]. However, [1] uses the Hadamard transform as an efficiently
computable random rotation of the observation, while our scheme uses the recursive structure of the Hadamard matrix
to construct an encoding scheme whose error decreases exponentially with the number of communication bits used.

**Reviewer 2:** *The $\varepsilon = \Omega(1)$ regime.* The reviewer mentions that the $\varepsilon = \Omega(1)$ regime is of practical interest. We agree
that this regime is frequently used in applications because of utility concerns, and in fact, this regime has become even
more relevant due to recent results in the shuffled model of DP [Erlingsson et al. (2020)].

*Related work and the presentation of tables.* We would like to point out that even though [9, 12] (Duchi-Jordan-
Wainwright) is not discussed in the "Related Work" section, we discuss their scheme `privUnit` in detail in Section A
and Section C, and compare its performance to our schemes. The experiments show that `privUnit`, while optimal
from a privacy perspective alone, does not perform well in the presence of communication constraints. In Table 2, the
blue (or red) color indicates that the scheme is optimal (or not). We will revise the tables to improve clarity.

*Tight upper bound for frequency estimation with $\ell_\infty$ loss.* We conjecture that our upper bound is tight, though this
requires proving a matching lower bound, which seems nontrivial. Note that [8] shows that the upper bound is tight for
$\varepsilon = O(1)$, but for a general $\varepsilon$, the problem remains open.

**Reviewer 3:** *Correctness of our main results.* The reviewer doubts the correctness of our main results due to the 1-bit
generic scheme in [8, Thm. 4.1]. First, we would like to clarify that the result in [8] works only for $\varepsilon = O(1)$, while in
practice $\varepsilon = \Omega(1)$ regime is also of great interest (see comment to R2). In our work, we show that 1 bit is not enough
for the $\varepsilon = \Omega(1)$ regime. Moreover, for larger $\varepsilon$ and $b$, it is not enough to repetitively apply the 1-bit scheme in [8]
(which is the same as "re-sampling"), since this makes the estimation error decay linearly in $b$ and $\varepsilon$. However, for
frequency estimation, we see that by cleverly designing the algorithm, we can achieve *exponential* decay with increasing
$b$, matching the lower bounds.

*How much public randomness is necessary?* It can be shown that our frequency estimation scheme (RHR) is optimal
not only in terms of the estimation error it achieves but also in terms of its use of public randomness. Recall that
for frequency estimation with $b$ bits communication and $\varepsilon$ LDP constraints, RHR uses $b^* \triangleq \min(b, \lceil \varepsilon \log_2 e \rceil)$ bits
communication and $\lceil \log d - b^* \rceil$ bits of public randomness, while [8] always uses $\lceil \log d \rceil$ bits of public randomness. By
slightly extending Theorem 4 in [1], one can show that at least $\log d - b^* - 2$ bits of public randomness is required to
get a consistent estimator. This implies that RHR is optimal in terms of the amount of shared randomness it needs, up to
an additive constant. For mean estimation, our scheme uses $\min(\lceil b^* \log d \rceil, d)$ bits of public randomness, while [8] uses
$d$ bits, which leads to a significant difference, e.g., in the case $b^* = 1$ considered in [8]. On the other hand, for discrete
distribution estimation (where $X_i \overset{\text{i.i.d.}}{\sim} \boldsymbol{p}$ with $\boldsymbol{p}$ supported on $\{1, ..., d\}$) our scheme requires no public randomness;
whereas [8] still needs it. The same conclusion holds for the distributional version of mean estimation (where $X_i \overset{\text{i.i.d.}}{\sim} P$
with $P$ supported on the Euclidean unit ball). As suggested by the reviewer, we will add this discussion about shared
randomness in our revised version.

**Reviewer 4:** We thank the reviewer for spotting the typos. We will fix the typos and proofread the revised version.

*Discussion on shared randomness.* For the more detailed discussion on shared randomness, please see the response
to R3. Indeed, for frequency estimation with communication budget $b < \log d - 2$, [1, Thm. 4] shows that shared
randomness is necessary, and we can further prove that our frequency estimation scheme RHR uses *the minimum*
*amount of shared randomness*. On the other hand, under distributional settings where local data is drawn from some
unknown but fixed distribution, our schemes for both frequency estimation and mean estimation require no shared
randomness.

[Meta-Review · NeurIPS 2020]

This paper studies several basic statistical estimation problems in a distributed setting subject to local differential privacy constraint. The reviewers agree that the results presented in the paper shed new light on the trade-off between privacy, accuracy, and communication. In particular, the paper provides results for a wider range of privacy loss regimes (epsilon). The reviewers reached consensus that this paper should be accepted.